# Visual Explanations of Image-Text Representations via Multi-Modal Information Bottleneck Attribution

**Ying Wang**[*]
New York University

**Tim G. J. Rudner**[*]
New York University

**Andrew Gordon Wilson**
New York University

## Abstract

Vision-language pretrained models have seen remarkable success, but their application to safety-critical settings is limited by their lack of interpretability. To improve the interpretability of vision-language models such as CLIP, we propose a multi-modal information bottleneck (M2IB) approach that learns latent representations that compress irrelevant information while preserving relevant visual and textual features. We demonstrate how M2IB can be applied to attribution analysis of vision-language pretrained models, increasing attribution accuracy and improving the interpretability of such models when applied to safety-critical domains such as healthcare. Crucially, unlike commonly used unimodal attribution methods, M2IB does not require ground truth labels, making it possible to audit representations of vision-language pretrained models when multiple modalities but no ground-truth data is available. Using CLIP as an example, we demonstrate the effectiveness of M2IB attribution and show that it outperforms gradient-based, perturbation-based, and attention-based attribution methods both qualitatively and quantitatively.

## 1 Introduction

Vision-Language Pretrained Models (VLPMs) have become the de facto standard for solving a broad range of vision-language problems [15]. They are pre-trained on large-scale multimodal data to learn complex associations between images and text and then fine-tuned on a given downstream task. For example, the widely used CLIP model [21], which uses image and text encoders trained on 400 million image-text pairs, has demonstrated remarkable performance when used for challenging vision-language tasks, such as Visual Question Answering and Visual Entailment [27].

VLPMs are highly overparameterized black-box models, enabling them to represent complex relationships in data. For example, Vision Transformers [ViTs; 7] are a state-of-the-art transformer-based vision model and are used as the image encoder of CLIP [21], containing 12 layers and 86 million parameters for ViT-Base and 24 layers and 307 million parameters for ViT-Large. Unfortunately, VLPMs like CLIP are difficult to interpret. However, model interpretability is essential in safety-critical real-world applications where VLPMs could be applied successfully, such as clinical decision-making or image captioning for the visually impaired. Improved interpretability and understanding of VLPMs would help us identify errors and unintended biases in VLP and improve the safety, reliability, and trustworthiness of VLPMs, thereby allowing us to deploy them in such safety-critical settings.

To tackle this lack of transparency in deep neural networks, attribution methods, which aim to explain a model's predictions by attributing contribution scores to each input feature, have been proposed for post-hoc interpretability. For example, for vision models, attribution methods can be used to create heatmaps that highlight features that are most responsible for a model's prediction. Similarly, for language models, scores are assigned to each input token. While the requirements for interpretability vary depending on the task, dataset, and model architecture, accurate and reliable attribution is an important tool for making emerging and state-of-the-art models more trustworthy.

---

[*]Equal contribution.

37th Conference on Neural Information Processing Systems (NeurIPS 2023).

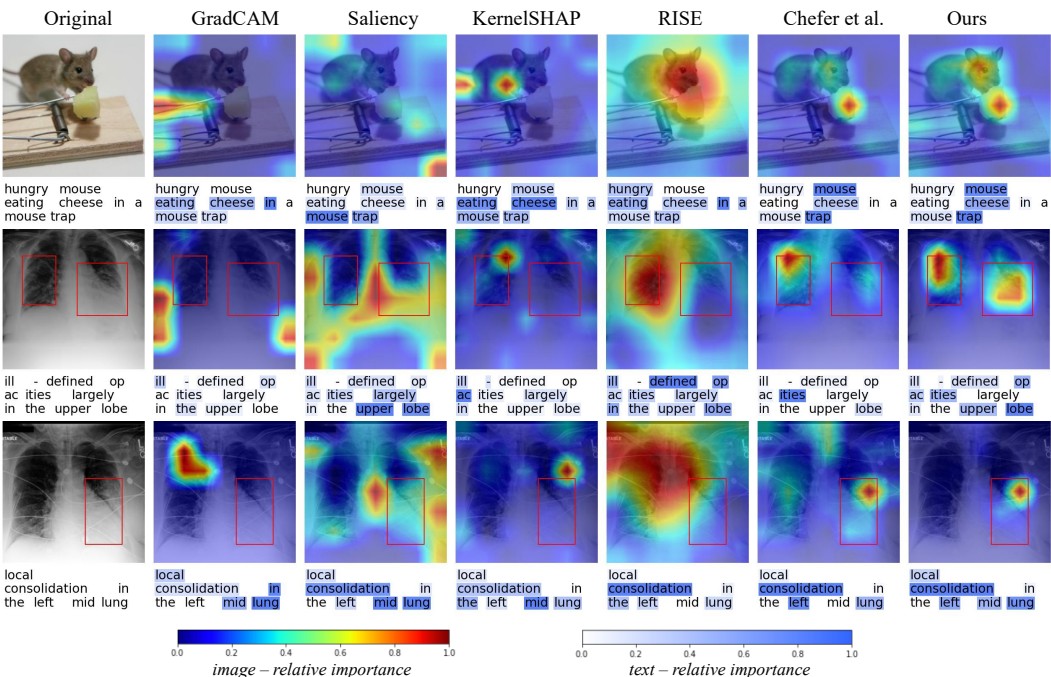

Figure 1: Example Attribution Maps For Image and Text Inputs. The red rectangles in the second and third rows show the ground-truth bounding boxes associated with the text, provided in the MS-CXR dataset [3]. Multi-modal information bottleneck (M2IB) attribution maps successfully identify relevant objects in the given image-text pairs, while other methods provide less precise localization and neglect critical features in the inputs.

While existing attribution methods focus primarily on unimodal models, we propose an attribution method for VLPMs that allows us to identify critical features in image and text inputs using the information bottleneck principle [31]. Unlike standard unimodal attribution methods, the proposed multi-modal information bottleneck (M2IB) attribution method *does not require access to ground-truth labels*.

To do this, we formulate a simple multi-modal information bottleneck principle, use a variational approximation to derive a tractable optimization objective from this principle, and optimize this objective with respect to a set of attribution parameters. Using these parameters, we are able to "turn off" all irrelevant features and only keep important ones and generate attribution maps. In contrast to unimodal information bottleneck attribution [24], we aim to find attribution parameters that maximize the likelihood of observing features of one modality given features associated with the respective other modality. We perform a qualitative and quantitative empirical evaluation and find that M2IB is able to successfully identify important features relevant to *both* image and text inputs. We provide an illustrative set of attribution map examples in Figure 1.

Our key contributions are summarized as follows:

1. We propose a multi-modal information bottleneck principle and use it to develop a multi-modal information bottleneck attribution method to improve the interpretability of Vision-Language Pretrained Models.
2. We perform an extensive empirical evaluation and demonstrate on several datasets—including healthcare data that can be used in safety-critical settings—that multi-modal information bottleneck attribution significantly outperforms existing gradient-based, perturbation-based, and attention-based attribution methods, quantitatively and qualitatively.

The code for our experiments is available at: https://github.com/YingWANGG/M2IB.

## 2 Related Work

Attribution methods enable post-hoc interpretability of trained models by allocating significance scores to the input features, such as image pixels or textual tokens.

### 2.1 Attribution methods

**Gradient-Based Attribution.** The use of gradients as a foundational component in attribution methods has been widely explored in the interpretability literature. Simonyan et al. [28] directly use the gradient of the target with respect to inputs as the attribution score. Building on this, Grad-CAM [Gradient-weighted Class Activation Mapping; 25] weights convolutional layer activations by average pixel-wise gradients to identify important regions in image inputs. Integrated Gradients [30] introduces two essential axioms for attribution methods— *sensitivity* and *implementation invariance*, which motivates the application of integrated gradients along paths from a baseline input to the instance under analysis.

**Perterbation-Based Attribution.** Perturbation approaches involve modifying input features to observe the resulting changes in model predictions. Unlike gradient-based techniques, they eliminate the need for backpropagation, allowing the model to be treated as a complete black box. However, such methods can be computationally intensive, especially for intricate model architectures, given the need to re-evaluate predictions under many perturbation conditions. The LIME [Local Interpretable Model-agnostic Explanations; 22] algorithm leverages perturbation in a local surrogate model. Lundberg and Lee [17] propose SHAP (SHapley Additive exPlanations), which employs a game-theoretic approach to attribution by utilizing perturbations to compute Shapley values for each feature. Furthermore, they propose a kernel-based estimation approach for Shapley values, called KernelSHAP, drawing inspiration from local surrogate models.

**Attention-based Attribution.** The rise of transformers across multiple domains in machine learning has necessitated the development of specialized attribution methods tailored to these models. Using attention as an attribution method offers a straightforward approach to deciphering and illustrating the significance of various inputs in a transformer's decision-making process. Nevertheless, solely relying on attention is inadequate as it overlooks crucial information from the value matrices and other network layers [5]. To address this issue, Chefer et al. [6] propose a method that learns relevancy maps through a forward pass across the attention layers with contributions from each layer cumulatively forming the aggregated relevance matrices.

**Information-Theoretic Attribution.** Schulz et al. [24] use information bottleneck attribution (IBA), where an information bottleneck is inserted into a layer of a trained neural network to distill the essential features for prediction. IBA is a model-agnostic method and shows impressive results on vision models including VGG-16 [14] and ResNet-50 [10]. Subsequently, IBA has been applied to language transformers [12] and was shown to also outperform other methods on this task. However, IBA has thus far been focused on only one modality and has only been adopted in supervised learning. To the best of our knowledge, there is no previous work on applying the information bottleneck principle to multi-modal models like VLPMs.

### 2.2 Evaluation of Attribution Methods

Despite active research on attribution methods, we still lack standardized evaluation metrics for attribution due to the task's inherent complexity. For vision models, attribution maps are frequently juxtaposed with ground-truth bounding boxes for a form of zero-shot detection. However, the attribution of high scores to irrelevant areas might arise from either subpar attribution techniques or flawed models, making it challenging to isolate the root cause of such discrepancies.

To tackle this issue, previous studies have resorted to degradation-based metrics [4, 33]. The underlying principle is that eliminating features with high attribution scores should diminish performance, whereas the removal of low-attributed features, often seen as noise, should potentially enhance performance. Additionally, Hooker et al. [11] introduce ROAR (Remove and Retrain), a methodology that deliberately degrades training and validation datasets in alignment with the attribution map, thereby accounting for potential distribution shifts. Should the attribution be precise, retraining the model using these altered datasets would result in a pronounced performance decline. Rong et al. [23] argue that mere image masking can cause a leak of information via the mask's shape, and thus propose a Noisy Linear Imputation strategy that replaces pixels with the average of their neighbors.

It is important to note that a visually appealing saliency map does not guarantee the efficacy of the underlying attribution method. For example, an edge detector might generate a seemingly plausible saliency map, yet it does not qualify as an attribution method because it is independent of the model under analysis. Adebayo et al. [1] propose a sanity check designed to assess whether the outputs of attribution methods genuinely reflect the properties of the specific model under evaluation. Notably, several prominent models, including Guided Backprop [29] and its variants appear insensitive to model weights and consequently do not pass the sanity check.

## 3 Attribution via a Multi-Modal Information Bottleneck Principle

In this section, we introduce a simple, multi-modal variant of the information bottleneck principle and explain how to adapt it to feature attribution.

### 3.1 The Information Bottleneck Principle

The information bottleneck principle [32] provides a framework for finding compressed representations of neural network models. To obtain latent representations that reflect the most relevant information of the input data, the information bottleneck principle seeks to find a stochastic latent representation $Z$ of the input source $X$ defined by a parametric encoder $p_{Z \mid X}(z \mid x; \theta)$ that is maximally informative about a target $Y$ while constraining the mutual information between the latent representation $Z$ and the input $X$. For a representation parameterized by parameters $\theta$, this principle can be expressed as the optimization problem

$$\max_\theta I(Z, Y; \theta) \quad \text{s.t.} \quad I(Z, X; \theta) \leq \bar{I}, \tag{1}$$

where $I(\cdot, \cdot; \theta)$ is the mutual information function and $\bar{I}$ is a compression constraint. We can equivalently express this optimization problem as maximizing the objective

$$\mathcal{F}(\theta) \doteq I(Z, Y; \theta) - \beta I(Z, X; \theta), \tag{2}$$

where $\beta$ is a Lagrange multiplier that trades off learning a latent representation that is maximally informative about the target $Y$ with learning a representation that is maximally compressive about the input $X$ [2].

### 3.2 A Multi-Modal Information Bottleneck Principle

Unfortunately, for VLPMs, the loss function above is not suitable since we wish to learn interpretable latent representations using only text and vision inputs without relying on task-specific targets $Y$ that may not be available or are expensive to obtain. To formulate a multi-modal information bottleneck principle for VLPMs, we need to develop an optimization objective that is more akin to optimization objectives for self-supervised methods for image-text representation learning that only use (text, image) pairs [21, 18].

This learning problem fundamentally differs from supervised attribution map learning for unimodal tasks. For example, we may have an image of a bear, $X_\text{bear}$, and a corresponding label, $Y_\text{bear} =$ "bear". For a unimodal classification task, we can simply maximize $I(Y_\text{bear}; Z_\text{bear}; \theta) - \beta I(X_\text{bear}; Z_\text{bear}; \theta)$ with respect to $\theta$, where $Z_\text{bear}$ is the latent representation of $X_\text{bear}$.

In contrast, in image-text representation learning, we typically have text descriptions, such as "This is a picture of a bear" ($L'_\text{bear}$) instead of labels [21]. In this setting, both $X_\text{bear}$ and $L'_\text{bear}$ are "inputs" without a pre-defined corresponding label. To obtain a task-agnostic image-text representation independent from any task-specific ground-truth labels, we would like to use both input modalities and define a multi-modal information bottleneck principle and whereas the outputs are closely dependent on the specific downstream task. This requires defining an alternative to the "fitting term" $I(Z, Y; \theta)$ of the conventional information bottleneck objective.

Fortunately, there is a natural proxy for the relevance of information in multi-modal data. If image and text inputs are related (e.g., text that describes the image), a good image encoding should contain information about the text and vice versa. Based on this intuition, we can express a multi-modal information bottleneck objective for $X_m$ with $m \in \mathcal{M} = \{\text{modality1}, \text{modality2}\}$, as

$$\mathcal{F}_m(\theta_m) = I(Z_m, E_{m'}; \theta_m) - \beta I(Z_m, X_m; \theta_m), \tag{3}$$

where $m' = \mathcal{M} \backslash m$ is the complement of $m$, and $E_{m'}$ is embedding of modality $m'$. Next, we will show how to use this multi-modal variant of the information bottleneck principle for attribution.

### 3.3 A Multi-Modal Information Bottleneck for Attribution

To compute attribution maps for image and text data without access to task-specific labels, we will define an information bottleneck attribution method for multi-modal data.

To restrict the information flow in the latent representation with a simple parametric encoder, we adapt the masking approach in Schulz et al. [24]. For clarity and brevity, we henceforth represent the $V_m \times W_m$-dimensional latent representation $Z_m$ in its vectorized form in $\mathbb{R}^J$ with $J \doteq V_m \cdot W_m$. Assuming independence across latent representation dimensions, we then define

$$p_{Z_m \mid X_m}(z_m \mid x_m; \theta_m) = \mathcal{N}(z_m; h_m(x_m; \lambda_m) \odot f_m^{\ell_m}(x_m), \sigma_m^2 \mathrm{diag}[(\mathbf{1}_J - h_m(x_m; \lambda_m))^2]), \quad (4)$$

where for a pair of modalities $\mathcal{M} \doteq \{\text{image}, \text{text}\}$ with $m \in \mathcal{M}$, $\theta_m \doteq \{\lambda_m, \sigma_m, \ell_m\}$ are parameters, $h_m(x_m; \lambda_m) \in \mathbb{R}^J$ is a mapping parameterized by $\lambda_m$, $f_m^{\ell}(X_m) \in \mathbb{R}^J$ is the vectorized output of the $\ell_m$th layer of modality-specific neural network embedding $f_m$, $\sigma_m^2 \in \mathbb{R}_{>0}$ is a hyperparameter, $\mathrm{diag}[\cdot]$ represents an operator that transforms a vector into a diagonal matrix by placing the vector's elements along the main diagonal, $\mathbf{1}_J \in \mathbb{R}^J$ is an all-ones vector, and $\odot$ is the Hadamard product. To avoid overloading notation, we will drop the subscript in probability density functions except when needed for clarity. Based on Equation (4), we can express the stochastic latent representations as

$$Z_m \mid x_m; \theta_m = h_m(x_m; \lambda_m) \odot f_m^{\ell_m}(x_m) + \sigma_m(\mathbf{1}_J - h_m(x_m; \lambda_m)) \odot \varepsilon, \quad (5)$$

where $\varepsilon \sim \mathcal{N}(0, I_J)$. From this reparameterization, we can see that $[h_m(x_m; \lambda_m)]_i = 1$ for $i \in \{1, ..., J\}$ means that no noise is added at index $i$, so $[Z_m]_i$ will be the same as the original $f_m^{\ell}(x_m)_i$, whereas $[h_m(x_m; \lambda_m)]_i = 0$ means that $[Z_m]_i$ will be pure noise.

We can now express the multi-modal information bottleneck attribution (M2IB) objectives as

$$\mathcal{F}_{\text{image}}(\theta_{\text{image}}) = I(Z_{\text{image}}, E_{\text{text}}; \theta_{\text{image}}) - \beta_{\text{image}} I(Z_{\text{image}}, X_{\text{image}}; \theta_{\text{image}}) \quad (6)$$

$$\mathcal{F}_{\text{text}}(\theta_{\text{text}}) = I(Z_{\text{text}}, E_{\text{image}}; \theta_{\text{text}}) - \beta_{\text{text}} I(Z_{\text{text}}, X_{\text{text}}; \theta_{\text{text}}), \quad (7)$$

which we can optimize with respect to the modality-specific sets of parameters $\lambda_{\text{image}}$ and $\lambda_{\text{text}}$, respectively. $\{\beta_{\text{image}}, \sigma_{\text{image}}, \ell_{\text{image}}\}$ and $\{\beta_{\text{text}}, \sigma_{\text{text}}, \ell_{\text{text}}\}$ are each sets of hyperparameters.

### 3.4 A Variational Objective for Multi-Modal Information Bottleneck Attribution

To obtain tractable optimization objectives, we use a variational approximation. First, we note that $I(Z_m, X_m; \theta_m) = \mathbb{E}_{p_{X_m}}[\mathbb{D}_{\text{KL}}(p_{Z_m \mid X_m}(\cdot \mid X_m; \theta_m) \parallel p_{Z_m}(\cdot; \theta_m))]$, where $Z_m \mid X_m; \theta_m$ can be sampled empirically whereas $p(z_m; \theta_m)$ does not have an analytic expression because the integral $p(z_m; \theta_m) = \int p(z_m \mid x_m; \theta_m) p(x_m) \, dx_m$ is intractable. To address this intractability, we approximate $p(z_m)$ by $q(z_m) \doteq \mathcal{N}(z_m; 0, I_J)$. This approximation leads to the upper bound

$$\begin{aligned} I(Z_m, X_m; \theta_m) &= \mathbb{E}_{p_{X_m}}[\mathbb{D}_{\text{KL}}(p_{Z_m \mid X_m}(\cdot \mid X_m; \theta_m) \parallel q_{Z_m}(\cdot))] - \mathbb{D}_{\text{KL}}(p_{Z_m} \parallel q_{Z_m}) \\ &\leq \mathbb{E}_{p_{X_m}}[\mathbb{D}_{\text{KL}}(p_{Z_m \mid X_m}(\cdot \mid X_m; \theta_m) \parallel q_{Z_m}(\cdot))] \\ &\doteq \mathcal{F}_m^{\text{compression}}(\theta_m). \end{aligned} \quad (8)$$

Next, while the unimodal information bottleneck attribution objective uses ground-truth labels to compute the "fit term" in the objective, the multi-modal information bottleneck attribution objectives require computing the mutual information between the aligned stochastic embeddings,

$$I(Z_m, E_{m'}; \theta_m) = \int p(e_{m'}, z_m; \theta_m) \log \frac{p(e_{m'}, z_m; \theta_m)}{p(e_{m'}) p(z_m)} \, de_{m'} \, dz_m \quad (9)$$

$$= \int p(e_{m'}, z_m; \theta_m) \log \frac{p(e_{m'} \mid z_m)}{p(e_{m'})} \, de_{m'} \, dz_m, \quad (10)$$

which is not in general tractable. To obtain an analytically tractable variational objective, we approximate the intractable $p(e_{m'} \mid z_m)$ by a variational distribution $q(e_{m'} \mid z_m) \doteq \mathcal{N}(e_{m'}; g_m(z_m), I_K)$, where $g_m$ is a mapping that aligns the latent representation of modality $m$ with $E_{m'}$ and $K$ is the dimension of the embedding, and get

$$\begin{aligned} I(Z_m, E_{m'}; \theta_m) &\geq \int p(e_{m'}, z_m; \theta_m) \log q(e_{m'} \mid z_m) \, de_{m'} \, dz_m \\ &= \int p(x_m) p(e_{m'} \mid x_m) p(z_m \mid x_m; \theta_m) \log q(e_{m'} \mid z_m) \, dx_m \, de_{m'} \, dz_m \\ &\doteq \mathcal{F}_m^{\text{fit}}(\theta_m). \end{aligned} \quad (11)$$

With this approximation, we can obtain a tractable variational optimization objective by sampling $X_m$ and $E_{m'}$ from the empirical distribution

$$\hat{p}(x_m, e_{m'}) \doteq \frac{1}{N} \sum_{n=1}^{N} \delta_0(x_m - x_m^{(n)}) \, \delta_0(e_{m'} - f_{m'}(x_{m'}^{(n)})), \tag{12}$$

where $f_{m'}(x_{m'})$ is the embedding input $x_{m'}$ under the VLPM. With these approximations, we can now state the full variational objective, which is given by

$$\mathcal{F}_m^{\text{approx}}(\theta_m) \doteq \mathcal{F}_m^{\text{fit}}(\theta_m) - \beta_m \mathcal{F}_m^{\text{compression}}(\theta_m). \tag{13}$$

The derivation of this objective has closely followed the steps in [2]. In practice, the objective can be computed using the empirical data distribution so that

$$\mathcal{F}_m^{\text{empirical}}(\theta_m) \doteq \frac{1}{N} \sum_{n=1}^{N} \int p(z_m \,|\, x_m^{(n)}; \theta_m) \log q(e_{m'} \,|\, z_m) \, \mathrm{d}z_m \\ - \beta_m \mathbb{D}_{\text{KL}}(p_{Z_m \,|\, X_m}(\cdot \,; x_m^{(n)}; \theta_m) \,\|\, q_{Z_m}(\cdot)). \tag{14}$$

**Final Variational Optimization Objective.** Finally, we assume that $h_m(x_m^{(n)}; \lambda_m) \doteq \lambda_m^{(n)}$ (i.e., each input point has its own set of attribution parameters), that $g_m$ is the mapping defined by the post-bottleneck layers of a VLPM for modality $m$, and that for each evaluation point the final embeddings *for each modality* get normalized across the embedding dimensions (i.e., both mapping, $g_m$ and $f_{m'}$, contain embedding normalization transformations). For normalized $g_m(z_m)$ and $e_{m'}$, the log of the Gaussian probability density $q(f_{m'}(x_{m'}) \,|\, g_m(z_m))$ simplifies and is proportional to the cosine similarity between $f_{m'}(x_{m'})$ and $g_m(z_m)$, giving the final optimization objective

$$\hat{\mathcal{F}}_m^{\text{empirical}}(\theta_m) \doteq \frac{1}{N} \sum_{n=1}^{N} \int p(z_m \,|\, x_m^{(n)}; \theta_m) S_{\text{cosine}}(e_{m'}, g_m(z_m)) \, \mathrm{d}z_m \\ - \beta_m \mathbb{D}_{\text{KL}}(p_{Z_m \,|\, X_m}(\cdot \,; x_m^{(n)}; \theta_m) \,\|\, q_{Z_m}(\cdot)), \tag{15}$$

where $S_{\text{cosine}}(\cdot, \cdot)$ is the cosine similarity function. For gradient estimation during optimization, the remaining integrals can be estimated using simple Monte Carlo estimation and reparameterization gradients. For $\theta_m = \{\lambda_m, \sigma_m, \ell_m\}$, the objective function is maximized with respect to $\lambda_m$ *independently for each modality*, and $\beta_m$, $\sigma_m$, and $\ell_m$ are modality-specifc hyperparameters.

## 4 Empirical Evaluation

We evaluate the proposed attribution method using CLIP [21] on four image-caption datasets, including widely-used image captioning datasets and medical datasets. Our main datasets are **(i)** Conceptual Captions [26] consisting of diverse images and captions from the web, and **(ii)** MS-CXR (Local Alignment Chest X-ray dataset; [3]), which contains chest X-rays and texts describing radiological findings, complementing MIMIC-CXR (MIMIC Chest X-ray; [13]) by improving the bounding boxes and captions. In addition, we also include some qualitative examples from the following radiology and remote sensory datasets to show the potential application of our model in safety-critical domains. Namely, we have **(iii)** ROCO (Radiology Objects in COntext; Pelka et al. [19]) that includes radiology image-caption pairs from the open-access biomedical literature database PubMed Central, and **(iv)** RSICD (Remote Sensing Image Captioning Dataset; Lu et al. [16]), which collects remote sensing images from web map services including Google Earth and provides corresponding captions.

### 4.1 Experiment Setup

For all experiments, we use a pretrained CLIP model with ViT-B/32 [7] as the image encoder and a 12-layer self-attention transformer as the text encoder. For Conceptual Captions, we use the pretrained weights of `openai/clip-vit-base-patch32`.[2] For MS-CXR, we use CXR-RePaiR [8] which is CLIP finetuned on radiology datasets, and compare the impact of finetuning in Section 4.5. For each {image, caption} pair, we insert an information bottleneck into the given layer of the text encoder and image encoder of CLIP separately, then train the bottleneck using the same setup as the *Per-Sample Bottleneck* of original IBA [24], which duplicates a single sample for 10 times to stabilize training and runs 10 iterations using the Adam optimizer with a learning rate of 1. Experiments show no significant difference between different learning rates and more training steps. We conduct a hyper-parameter tuning on the index of the layer $l$, the scaling factor $\beta$, and the variance $\sigma^2$. For a discussion of these hyperparameters in the multi-modal information bottleneck objective, see Appendix A.

---

[2]Cf. `https://huggingface.co/openai/clip-vit-base-patch32`.

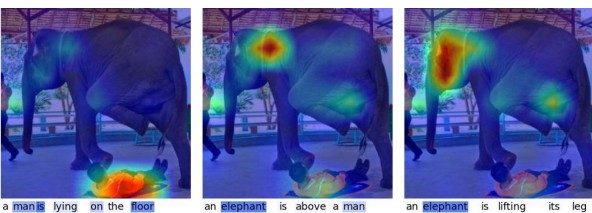 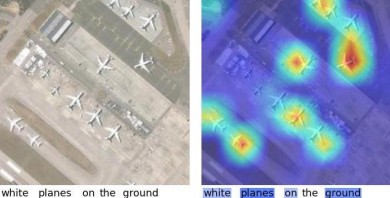

| (a) Different Highlights with Respect to Different Text. | (b) Multiple Occurrences of Same Object. |

Figure 2: Example Saliency Maps when Involving Multiple Objects. Our method can successfully detect all occurrences of all relevant objects in image and text.

## 4.2 Qualitative Results

We qualitatively compare our method with 5 widely used attribution methods, including gradient-based GradCAM [25] and Saliency method [28], perturbation-based Kernel SHAP [17] and RISE [20], and transformer-specific method [6]. We show one image-caption example and two radiology examples in Figure 1 and provide more qualitative comparisons in the Appendix B. As shown, our method is able to capture all relevant objects appearing in both modalities, while other methods tend to focus on one major object. As illustrated in Figure 2a, the highlighted areas in the image change according to different inputs, and our method can capture complicated relationships between image and text when involving multiple objects. Our method is also able to detect multiple occurrences of the same object in the image, as illustrated in Figure 2b.

## 4.3 Localization Test

We quantitatively measure the effectiveness of our proposed attribution method by evaluating its accuracy in zero-shot detection for images. We binarize the saliency map such that the areas with scores higher than the threshold (75%) are assigned 1 while the rest are assigned 0. We denote the resulting binary map as $S_{\mathrm{pred}}$. We also construct a ground-truth binary map, $S_{\mathrm{gt}}$, using the bounding boxes provided by MS-CXR [3], where the region inside the bounding boxes is assigned to 1 while the outside is assigned to 0. Note that some samples have multiple bounding boxes, and we consider all of them to test the method's multi-occurrence detection ability. Then, we calculate the IoU (Intersection over Union) of $S_{\mathrm{pred}}$ and $S_{\mathrm{gt}}$. Namely, for images with a height of $n$ and a width of $m$, the score is calculated by

$$\text{Localization} = \frac{\sum_{i=1}^{n} \sum_{j=1}^{m} \mathbb{1}_{S_{\mathrm{pred}}^{ij} \wedge S_{\mathrm{gt}}^{ij}}}{\sum_{i=1}^{n} \sum_{j=1}^{m} \mathbb{1}_{S_{\mathrm{pred}}^{ij} \vee S_{\mathrm{gt}}^{ij}}}, \tag{16}$$

where $\mathbb{1}$ is the indicator function, $\wedge$ is the logical AND and $\vee$ is the logical OR operator.

We found that M2IB attribution attains an average IoU of 22.59% for this zero-shot detection task, outperforming all baseline models, as shown in Table 1. Recognizing that a localization score of 22.59% appears somewhat low in absolute terms, we briefly note that there are two potential causes that could lead to the low absolute values in the localization scores: **(i)** M2IB attribution indeed generates segmentation instead of bounding boxes, so evaluation by bounding boxes would underestimate the quality of the saliency map. **(ii)** The model under evaluation [8] is not finetuned for detection and may only have learned a coarse-grained relationship between X-rays and medical notes.

## 4.4 Degradation Tests

While the localization test shows that M2IB attribution may be a promising zero-shot detection and segmentation tool, the localization test may underestimate the accuracy of attribution since even a perfect attribution map can produce a low localization score if the (finetuned) VLPM under evaluation is poor at extracting useful information—which is very likely for challenging specialized tasks like chest X-ray classification.

To get a more fine-grained picture of the usefulness of M2IB, we use three additional evaluation metrics to compare M2IB to competitive baselines. The underlying idea of all three evaluations is that removing features with high attribution scores should decrease the performance, while discarding features with low attribution scores can improve the performance as noisy information is ignored. We randomly sample 2,000 image-text pairs from Conceptual Captions and 500 image-text pairs from MS-CXR. For each dataset, we perform ten experiments for each evaluation metric (five for ROAR+) and report the average score with the standard error in Table 1.

Table 1: Quantitative Results. The boldface denotes the best result per row. Means and standard errors were computed over ten random seeds.

| | Methods | GradCAM [25] | Saliency [28] | KS [17] | RISE [20] | Chefer et al. [6] | Ours |
|---|---|---|---|---|---|---|---|
| CC image | % Conf. Drop ↓ | 4.96 ± 0.01 | 1.99 ± 0.01 | 1.94 ± 0.01 | 1.12 ± 0.01 | 1.63 ± 0.01 | **1.11** ± 0.01 |
| | % Conf. Incr. ↑ | 17.84 ± 0.08 | 22.95 ± 0.12 | 25.18 ± 0.28 | 35.72 ± 0.14 | 37.41 ± 0.12 | **41.55** ± 0.19 |
| | % ROAR+ ↑ | 2.29 ± 0.41 | 6.88 ± 0.88 | 1.56 ± 0.88 | 3.15 ± 0.97 | 7.66 ± 0.55 | **10.59** ± 0.85 |
| CC text | % Conf. Drop ↓ | 2.19 ± 0.01 | 1.78 ± 0.01 | 1.71 ± 0.01 | 1.30 ± 0.01 | **1.06** ± 0.01 | **1.06** ± 0.01 |
| | % Conf. Incr ↑ | 29.71 ± 0.19 | 38.96 ± 0.15 | **46.87** ± 0.21 | 38.31 ± 0.48 | 38.42 ± 0.11 | 38.55 ± 0.20 |
| | % ROAR+ ↑ | 43.23 ± 0.66 | 43.74 ± 0.65 | 47.46 ± 3.62 | 49.04 ± 1.12 | 53.57 ± 1.26 | **60.41** ± 1.12 |
| MSCXR image | % Conf. Drop ↓ | 2.76 ± 0.03 | 0.81 ± 0.01 | 2.37 ± 0.04 | 3.94 ± 0.03 | 1.87 ± 0.02 | **0.55** ± 0.01 |
| | % Conf. Incr. ↑ | 12.64 ± 0.46 | 35.08 ± 0.44 | 10.24 ± 0.68 | 7.28 ± 0.44 | 21.44 ± 0.46 | **45.92** ± 0.70 |
| | % ROAR+ ↑ | 3.54 ± 0.80 | 25.46 ± 1.35 | 12.67 ± 1.02 | 16.79 ± 0.76 | 24.42 ± 1.19 | **38.7** ± 0.86 |
| | % Localization ↑ | 5.56 ± 0.13 | 21.6 ± 0.16 | 7.77 ± 0.13 | 10.97 ± 0.24 | 21.65 ± 0.25 | **22.59** ± 0.14 |
| MSCXR text | % Conf. Drop ↓ | 2.26 ± 0.04 | 3.35 ± 0.03 | 2.4 ± 0.05 | **1.16** ± 0.02 | 2.93 ± 0.03 | 2.28 ± 0.04 |
| | % Conf. Incr. ↑ | 36.24 ± 0.54 | 18.88 ± 0.54 | 34.12 ± 0.77 | **57.2** ± 0.65 | 28.08 ± 0.34 | 35.48 ± 0.69 |
| | % ROAR+ ↑ | 11.07 ± 0.62 | 15.79 ± 0.92 | 14.28 ± 1.09 | 12.09 ± 1.52 | 9.11 ± 0.6 | **16.31** ± 0.75 |

| Original | Saliency Map | Attribution Weighted Image | Corrupted (binarized M) |
|---|---|---|---|

a man holds what is believed to be some of the debris that caused damage to vehicles .

a man holds what is believed to be some of the debris that caused damage to vehicles !

\<B\> man holds \<B\> \<B\> \<B\> to \<B\> \<B\> \<B\> \<B\> debris that \<B\> damage \<B\> vehicles .

a \<B\> \<B\> what is believed \<B\> be some of the \<B\> \<B\> caused \<B\> to \<B\> \<B\>

Figure 3: Visualization of Degradation. The third column is obtained by calculating the element-wise product of the original image and saliency map, while the text with attribution scores lower than 50% percentile is masked by a blank token . It is used in the *Increase in Confidence* metric and *Drop in Confidence* metric. The fourth column shows an example of the training data in ROAR+. We replace the image pixels with attribution scores higher than 75% percentile by the channel mean and replace the text tokens with attribution scores higher than 50% by a blank token . The results in Table 1 use the padding token as the blank token .

**Drop in Confidence** [4]. An ideal attribution method should only assign high scores to important features, thus we should not observe a drop in performance if only the high-attribution parts are allowed in the input. For images, we use point-wise multiplication of the saliency map and the image input. Since scaling token ids is meaningless, we use binarization similar to [33] where only tokens with attribution scores in the top 50% are kept. We provide an example of distilled image and text in Figure 3. Formally, we define this score by

$$\text{Confidence Drop} = \frac{1}{N} \sum_{i=1}^{N} \max(0, o_i - s_i),$$ (17)

where $o_i$ is the cosine similarity of features of original images and texts, and $s_i$ is the new cosine similarity when one modality is distilled according to the attribution. The lower this metric is, the better the attribution method is. This metric is implemented in the `pytorch-gradcam` repository.[3]

**Increase in Confidence** [4]. Similarly, removing noisy information in the input might increase the model's confidence. We compute

$$\text{Confidence Increase} = \frac{1}{N} \sum_{i=1}^{N} \mathbb{1}(o_i < s_i),$$ (18)

where $\mathbb{1}$ is the indicator function and the definition of $o_i$ and $s_i$ is the same as above. Higher values indicate better performance. This metric is also implemented in the `pytorch-gradcam` repository.[3]

---

[3]GradCAM and its variants are usually applied to image classification and use the softmax outputs of each class as confidence scores. Since our setting does not contain any labels, we use cosine similarity with the other modality instead.

**Remove and Retrain +** (ROAR+, an Extension of ROAR [11]). We finetune the base model on the degraded images and texts where the most important parts are replaced by uninformative values (i.e., channel means of images or padding tokens for texts, see Figure 3) and evaluated on a validation set of original inputs.[4] If the attribution method is accurate, a sharp decrease in performance is expected because all useful features are removed, and the model cannot learn anything relevant from the degraded data. We split the testing dataset into 80% training data and 20% validation data. We use the same contrastive loss as used for pretraining CLIP and define the score by $(l_c - l_o)/l_o$, where $l_o$ is the validation loss of retraining using the original data, and $l_c$ is the validation loss when retraining with corrupted data. We repeat the process five times and report the average score.

The results are summarized in Table 1. M2IB attribution outperforms baseline models in almost all numerical metrics, except for perturbation-based methods, which achieve better Increase/Drop in Confidence scores for texts. Perturbation-based methods perform better for short text because they can scan all possible binary masks on the text and then find the optimal one with the highest confidence score. However, this kind of method is very computationally expensive. We use 2,048 masks for image and 256 masks for text in RISE, where each mask of each modality requires one forward pass to get the confidence score, resulting in 2.3k forward passes (7.8s on RTX8000 with a batch size of 256). In contrast, M2IB attribution only requires 100 forward passes and takes 1.2s for one image-text pair.

In general, removing pixels or tokens with lower attribution scores using M2IB attribution generally increases the mutual information with the other modality, while masking by our attribution map generally decreases the relevance with the other modality and makes the model perform worse when retraining on the corrupted data. This confirms that our method generates useful attribution maps.

## 4.5 Sanity Check

We conduct a sanity check to ensure our method is, in fact, sensitive to model parameters. We follow the sanity check procedure proposed by Adebayo et al. [1], where parameters in the model are randomized starting from the last to the first layer. As shown in Figure 4, M2IB passes the sanity check as the attribution scores of image pixels and text tokens change as the model weights change. Our method also produces more accurate saliency maps for finetuned models compared to pretrained models, which further confirms that the resulting attribution can successfully reflect the quality of the model. Since we insert the information bottleneck after a selected layer (layer 9 in this case), the randomization of this and previous layers appears to have a larger influence on the output.

## 4.6 Error Analysis and Limitations

We notice that our proposed attribution method generally performs well on text but sometimes shows less satisfying performance on images. By inspecting the qualitative examples, we observe that M2IB sometimes fails to detect the entire relevant regions in images. As shown in the fourth ("sea" example) and fifth ("bridge" example) rows in Figure 7, our method only highlights a fraction of the object in the image, although it should include the whole object. This is probably due to the fact that the model under evaluation only relies on a few patterns in the image to make its prediction. Increasing the relative importance of the fitting term (i.e., using smaller $\beta$) enlarges the highlighted area. However, we don't recommend using extreme $\beta$ because it will break the balance between fitting and compression and thus make the information bottleneck unable to squeeze information.

We also note that M2IB is sensitive to the choice of hyperparameters. As shown in Figure 5, different combinations of hyperparameters will generate different saliency maps. We show how to use the ROAR+ score to systematically select the optimal hyperparameters and also provide visualization to illustrate the effect of different hyperparameters in Appendix A. Since there is no convention on evaluating the attribution method, we suggest considering various evaluation metrics, visualization of examples, and the goal of the attribution task when choosing hyperparameters. We emphasize that M2IB should be used with caution since attributing the success or failure of a model solely to a set of features can be overly simplistic, and different attribution methods might lead to different results.

---

[4]The original ROAR also corrupts the validation data, which means that different methods might use varied training and validation datasets, making direct comparisons difficult. To ensure a fair comparison, we consistently use the original data as the validation set for all methods.

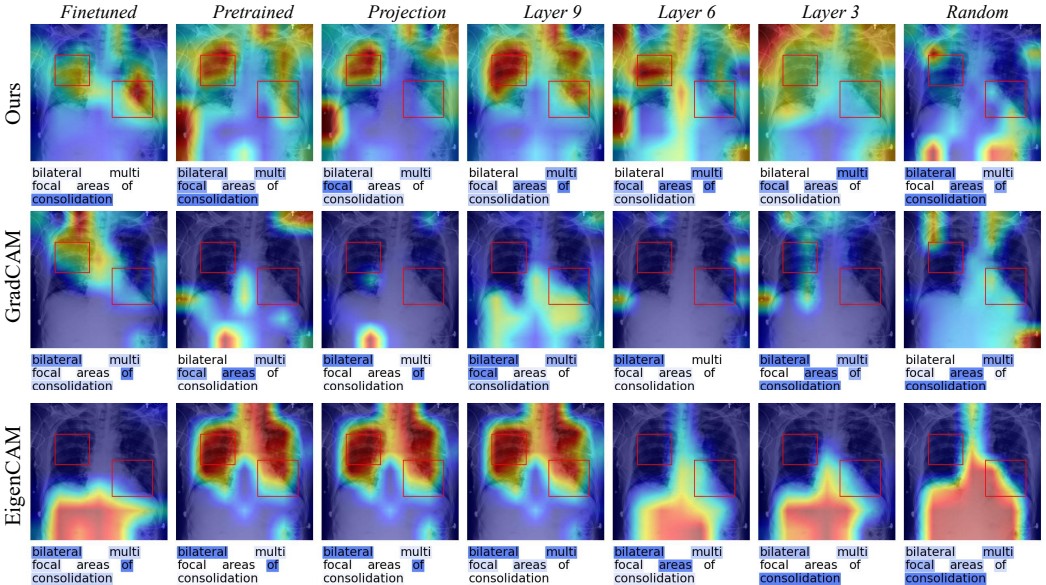

Figure 4: Saliency Maps for Sanity Checks. "Finetuned" represents the model that is finetuned on MIMIC-CXR [13], a Chest X-ray dataset. "Pretrained" represents pretrained CLIP [21] from OpenAI. "Projection" represents CLIP with randomized projection layer, which is the last layer of the image encoder or text encoder that projects image or text features into the shared embedding space. "Random" means that all parameters in the model are randomly initiated. The remaining columns represent models with weights randomized starting from the last to the given layer. The results suggest that the saliency maps of M2IB attribution are sensitive to model weights, as desired, meaning that M2IB passes the sanity check.

## 5  Discussion

We developed multi-modal information bottleneck (M2IB) attribution, an information-theoretic approach to multi-modal attribution mapping. We used CLIP-ViT-B/32 in our experiments, but M2IB attribution can be directly applied to CLIP with alternative neural network architectures and is compatible with any VLPM for which the features of all modalities are projected into a shared embedding space—a commonly used approach in state-of-the-art multi-modal models.

Going beyond vision and language modalities, Girdhar et al. [9] recently proposed a new multi-modal model, ImageBind, which aligns embeddings of five modalities to image embeddings through contrastive learning on pairs of images with each modality and uses a similar architecture as CLIP, where the encoder for each modality is a transformer. We applied M2IB attribution to pairs of six different modalities using ImageBind—which required minimal implementation steps—and provided qualitative examples in Appendix C to illustrate that M2IB attribution can be applied successfully to more than just vision and text modalities.

In this paper, we provided exhaustive empirical evidence that M2IB increases attribution accuracy and improves the interpretability of state-of-the-art vision-language pretrained models. We hope that this work will encourage future research into multi-modal information-theoretic attribution methods that can help improve the interpretability and trustworthiness of VLPMs and allow them to be applied to safety-critical domains where interpretability is essential.

## Acknowledgments

This work was supported in part through the NYU High-Performance Computing services as well as by NSF CAREER IIS-2145492, NSF I-DISRE 193471, NSF IIS-1910266, BigHat Biosciences, Capital One, and an Amazon Research Award.

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

# Appendix

**Table of Contents**

## Appendix A    Experimental Details and Further Experimental Results

$\beta$ controls the relative importance of the compression term. The larger the $\beta$ is, the less information is allowed to flow through this layer. As shown in Figure 5a, too large and too small $\beta$ generates similar attribution maps in terms of relative importance. However, too small $\beta$ allows nearly everything through the bottleneck, whereas too large $\beta$ nearly discards everything.

$\sigma$ controls the values of the noise added to the intermediate representations. When $\sigma$ is very small, the values of the noise will be close to 0, thus having minimal impact on the intermediate representations. This effect is similar to the situation when $\beta$ is very small and IB will add almost no noise (Figure 5b). $\sigma$ also directly affects the compression term as smaller $\sigma$ will lead to higher KL divergence. Thus, $\sigma$ and $\beta$ are correlated with each other and we perform a grid search to find the best combination.

Layer $\ell$ where the information bottleneck is inserted also impacts the attribution. Inserting the bottleneck too early will prevent the model from learning informative features while inserting the bottleneck too late reduces the impact of the bottleneck (Figure 5c). We also observe that the attribution of texts is usually more stable than images.

These hyperparameters can be chosen according to numerical metrics mentioned in Section 4.4. We randomly sample 500 image-text pairs from Conceptual Captions and 500 image-text pairs from MS-CXR, which is ensured to not overlap with the test set in Section 4. We then perform a grid search for the best combination of $\beta = \{1, 0.1, 0.01\}$ and $\sigma^2 = \{1, 0.1, 0.01\}$ and $\ell = \{7, 8, 9\}$ (indexing from 0), and find the best combination of hyperparameters is shown in Table 2.

Table 2: Hyperparameter Tuning Results for $\ell$, $\beta$ and $\sigma$: We calculate the ROAR+ score for different combinations for 3 runs and report the average. For each table, the highest score is in bold and indicates the performance is optimal for this set of hyperparameters. For texts of Conceptual Captions, we select $\ell = 9$, $\beta = 0.1$ and $\sigma = 1$. Since ROAR+ uses binarized saliency maps (75% threshold for image pixels and 50% threshold for text tokens, it mainly focuses on the features with high attribution scores and neglects the change in the attribution for less important features. Thus, sometimes hyperparameters with slightly lower ROAR+ scores might generate more visually appealing results as in Figure 5, though the numerical results are consistent with qualitative examples in general.

(a) Conceptual Captions - Image

|  |  | $\sigma^2=1$ | $\sigma^2=0.1$ | $\sigma^2=0.01$ |
|---|---|---|---|---|
| $\ell=7$ | $\beta=1$ | 10.13 | 10.78 | 7.11 |
|  | $\beta=0.1$ | 8.00 | 7.96 | 6.66 |
|  | $\beta=0.01$ | 7.12 | 6.69 | 8.32 |
| $\ell=8$ | $\beta=1$ | 12.08 | 8.25 | 9.49 |
|  | $\beta=0.1$ | 10.33 | 6.96 | 6.33 |
|  | $\beta=0.01$ | 10.05 | 6.77 | 8.23 |
| $\ell=9$ | $\beta=1$ | 7.75 | 9.44 | 5.63 |
|  | $\beta=0.1$ | **12.72** | 8.91 | 9.94 |
|  | $\beta=0.01$ | 8.88 | 6.42 | 9.08 |

(b) Conceptual Captions - Text

|  |  | $\sigma^2=1$ | $\sigma^2=0.1$ | $\sigma^2=0.01$ |
|---|---|---|---|---|
| $\ell=7$ | $\beta=1$ | 15.49 | 10.74 | 10.78 |
|  | $\beta=0.1$ | 13.31 | 13.73 | 14.18 |
|  | $\beta=0.01$ | 12.96 | 15.61 | 12.78 |
| $\ell=8$ | $\beta=1$ | 11.29 | 16.47 | 8.82 |
|  | $\beta=0.1$ | 13.26 | 14.80 | 13.86 |
|  | $\beta=0.01$ | 8.99 | 11.09 | 13.72 |
| $\ell=9$ | $\beta=1$ | 13.85 | **18.16** | 9.87 |
|  | $\beta=0.1$ | **18.16** | 15.03 | 8.97 |
|  | $\beta=0.01$ | 11.72 | 8.46 | 9.80 |

(c) MSCXR - Image

|  |  | $\sigma^2=1$ | $\sigma^2=0.1$ | $\sigma^2=0.01$ |
|---|---|---|---|---|
| $\ell=7$ | $\beta=1$ | 35.24 | 29.56 | 25.20 |
|  | $\beta=0.1$ | 34.15 | 30.44 | 30.28 |
|  | $\beta=0.01$ | 30.12 | 30.21 | 29.32 |
| $\ell=8$ | $\beta=1$ | 34.59 | 29.35 | 30.07 |
|  | $\beta=0.1$ | 32.33 | 30.68 | 28.04 |
|  | $\beta=0.01$ | 31.99 | 33.24 | 36.99 |
| $\ell=9$ | $\beta=1$ | 31.76 | 28.43 | 32.94 |
|  | $\beta=0.1$ | **37.17** | 35.06 | 32.22 |
|  | $\beta=0.01$ | 36.19 | 34.59 | 30.18 |

(d) MSCXR - Text

|  |  | $\sigma^2=1$ | $\sigma^2=0.1$ | $\sigma^2=0.01$ |
|---|---|---|---|---|
| $\ell=7$ | $\beta=1$ | **15.77** | 10.91 | 9.22 |
|  | $\beta=0.1$ | 12.60 | 11.15 | 8.92 |
|  | $\beta=0.01$ | 13.89 | 12.31 | 13.11 |
| $\ell=8$ | $\beta=1$ | 10.76 | 14.34 | 12.13 |
|  | $\beta=0.1$ | 12.67 | 10.53 | 11.09 |
|  | $\beta=0.01$ | 10.58 | 12.49 | 15.48 |
| $\ell=9$ | $\beta=1$ | 11.70 | 11.01 | 12.24 |
|  | $\beta=0.1$ | 10.39 | 13.54 | 12.48 |
|  | $\beta=0.01$ | 12.93 | 14.64 | 12.57 |

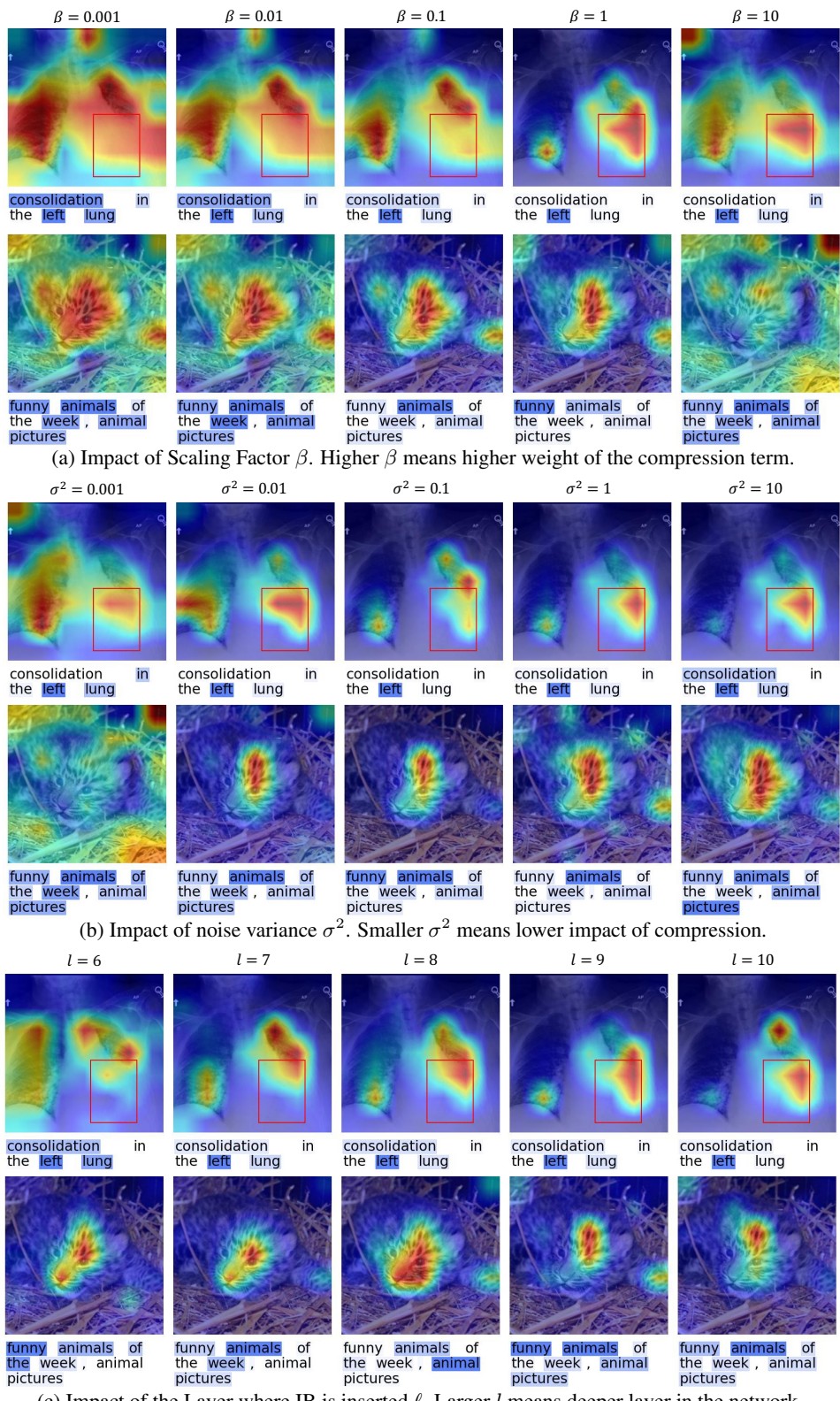

(a) Impact of Scaling Factor $\beta$. Higher $\beta$ means higher weight of the compression term.

(b) Impact of noise variance $\sigma^2$. Smaller $\sigma^2$ means lower impact of compression.

(c) Impact of the Layer where IB is inserted $\ell$. Larger $l$ means deeper layer in the network.

Figure 5: Visualization of the Impact of Different Hyperparameters. $\beta$ and $\sigma^2$ that make the fitting and compression terms be at a similar scale and deeper layer $\ell$ usually give better performance.

# Appendix B    Attribution Maps for Additional Examples

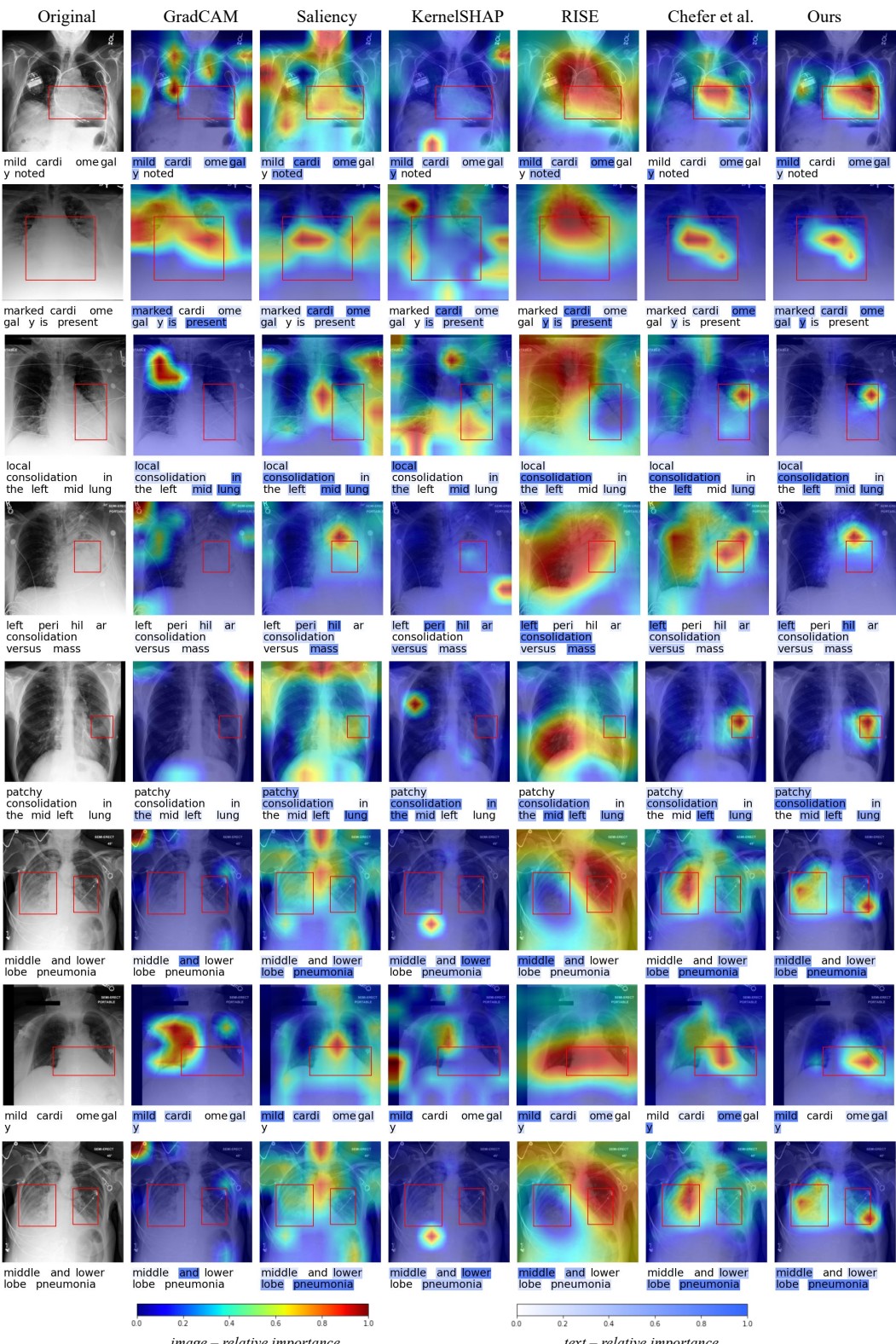

Figure 6: Attribution maps for randomly picked examples from the MS-CXR chest X-ray dataset [3]. Note that the attribution score is assigned to each token, instead of each word, due to tokenization.

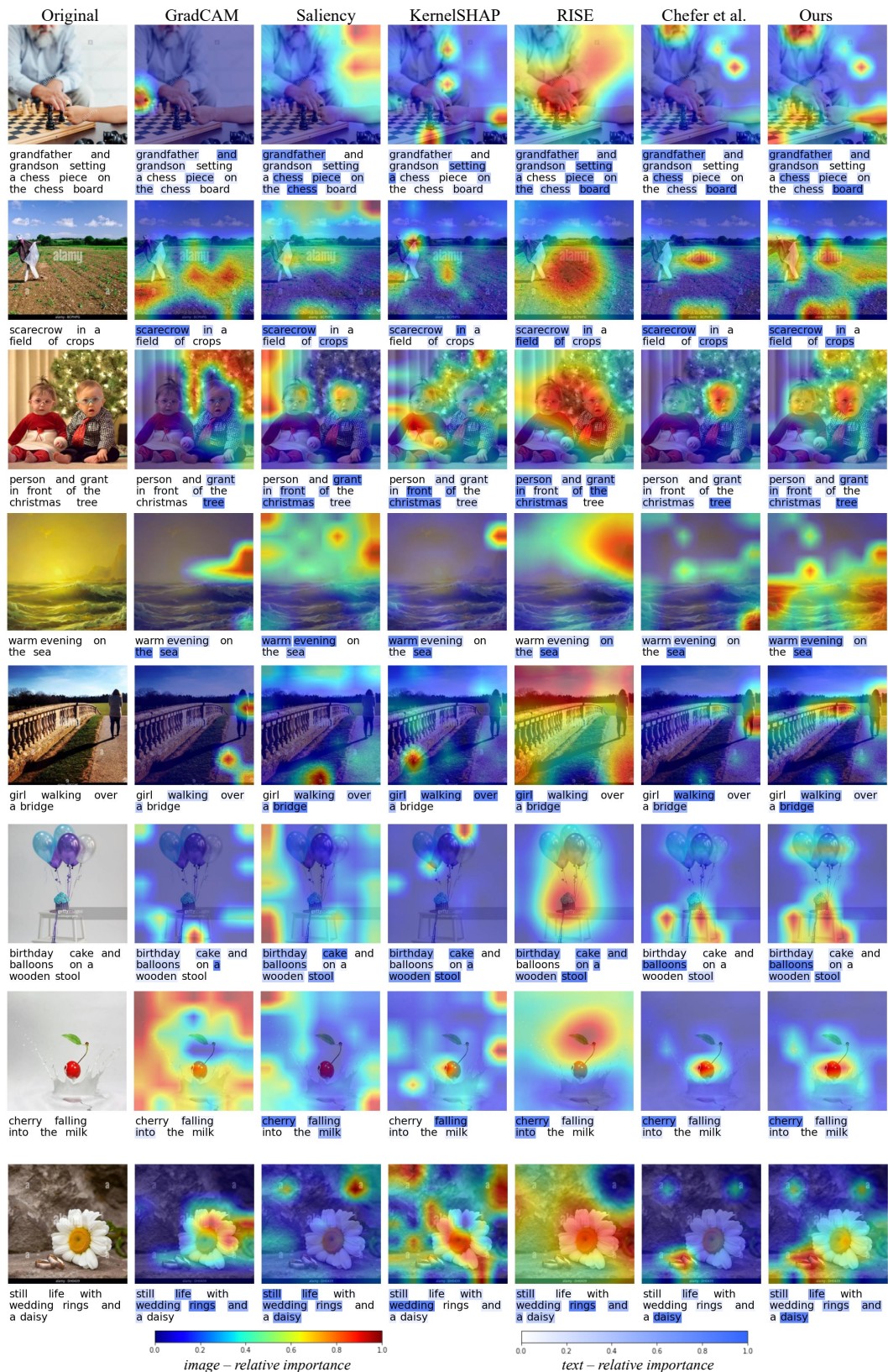

Figure 7: Attribution maps for randomly picked examples from the Conceptual Captions dataset [26].

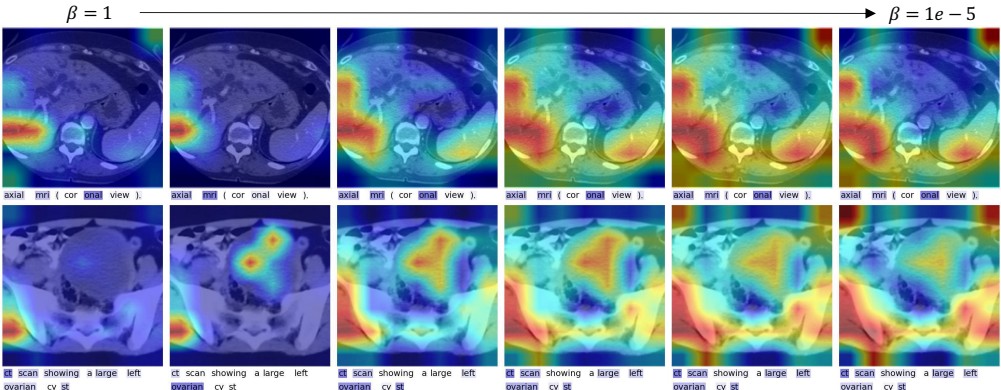

$\beta = 1$ —————————————→ $\beta = 1e - 5$

Figure 8: Attribution maps for MRI and CT examples from the ROCO dataset [19]. The highlighted areas increase when $\beta$ decreases. Due to the lack of segmentation masks in ROCO, we are unable to perform a quantitative evaluation.

## Appendix C    Extension to Additional Modalities With ImageBind

Our method can be easily extended to representation learning of modalities other than image and text, as long as features of different modalities are projected into a shared embedding space. We use ImageBind [9] as an example to showcase the effectiveness of our method in interpreting audio-image and audio-text representation learning. Since ImageBind aligns other modalities (audio, sensors that record depth (3D), thermal (infrared radiation), and inertial measurement units (IMU)) embedding to image embeddings through contrastive learning, the embeddings of all modalities are in one shared embedding space. Thus, we can insert the information bottleneck into certain layers of the feature encoders of ImageBind respectively and adopt the M2IB objective similar to Equation (6) and Equation (7).

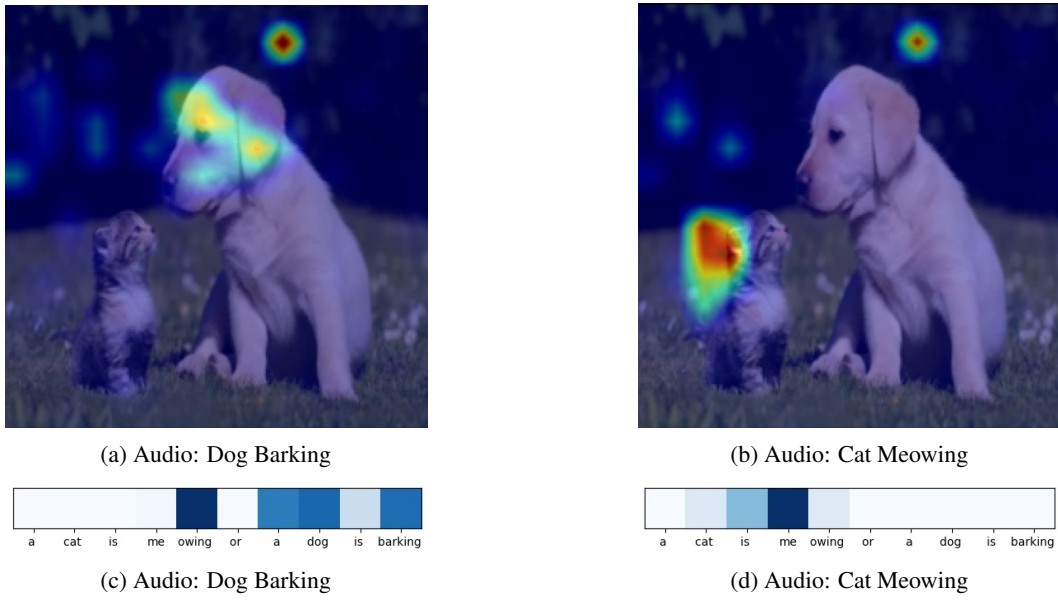

(a) Audio: Dog Barking     (b) Audio: Cat Meowing

(c) Audio: Dog Barking     (d) Audio: Cat Meowing

Figure 9: Attribution map on image and text when the other modality is audio.

