# OpenReview forum: "Visual Explanations of Image-Text Representations via Multi-Modal Information Bottleneck Attribution"
_NeurIPS.cc/2023/Conference — NeurIPS 2023 poster_

### Official Review · Reviewer_6SYf · 2023-06-27

**Soundness:** 3 good
**Presentation:** 3 good
**Contribution:** 2 fair
**Rating:** 5
**Confidence:** 5

**Summary:**

Visual-language models (VLMs) are increasingly used for multiple vision-language problems like object recognition, image classification, image captioning, etc. Formally, these models are trained in a self-supervised fashion using large datasets comprising millions of image-text pairs, such that they learn generic image and text representations during the training, which can be fine-tuned for different downstream tasks. However, VLMs are intrinsically very complex and are hard to interpret. In this work, the authors use CLIP, a state-of-the-art visual-language model, and propose a multi-modal information bottleneck (M2IB) objective that compresses irrelevant and noisy features and retains visual and text features relevant to the underlying VLM without requiring ground truth labels. Further, the authors show the effectiveness of M2IB in generating attribution maps with increased attribution accuracy across several high-stakes datasets, such as medical diagnosis.

**Strengths:**

1. One of the main highlights of the work is that it does not require the ground-truth label for generating attribution maps and approximates its impact using mutual information between the image and text representations.

2. In contrast to existing works that have focused on a single modality (images or texts) for supervised learning, the authors propose applying the information bottleneck principle to aid our understanding of the inner mechanisms of VLMs using both image and textual inputs.

3. The proposed solution is clear and appears to be a simple extension of the information bottleneck paper referred to in the paper.

**Weaknesses:**

1. The hypothesis that a good image encoding should contain information about the text, while a good text encoding should include information about the image is unclear. This is fundamentally questionable as the model never learns a unique mapping between the **Z_image** and **Z_test** representations, i.e., the representations in VLMs are learned in a compact manner and it may be difficult to disentangle representations between similar concepts, say **Z_bear_image** and **Z_panda_text**.

2. The proposed technique comprises a large set of hyperparameters for tuning M2IB and generating attribution maps as explanations and the authors do not provide a clear analysis of how to tune or fix them for different models and datasets.

3. In Lines 214-216, the authors mention that "for each {image, caption} pair, we insert an information bottleneck into the given layer of the text encoder and image encoder of CLIP separately, then train the bottleneck using the same setup as the Per-Sample Bottleneck of original IBA". This sounds overly computationally expensive as we need to tune all the hyperparameters for each <image, caption> pair!

4. In Lines 228-229, the authors state that "our method is able to capture all relevant objects appearing in both modalities, while other methods tend to focus on one major object" --- is that the aim of the results? It is totally possible that the model is only using limited information to learn a particular image-caption pair. I can see why it's logically unintuitive to humans, but that can totally be what the model is actually doing. A good check would be how different the mutual information on using the attribution generated using CAM-based methods and the proposed method.

5. Minor comment: the paper has several typos and the writing can be improved. For example, "dind" in Line 54, "emoirical" in Line 183, missing references CXR-RePaiR [] in Line 249, and "test encoder of CLIP" in Figure 4 caption. Also, the paper discusses a lot about future works in Section 5. It would be better to talk about the limitations of the work instead.

**Questions:**

1. The authors mention that they use a "variational approximation upper bound to estimate the information in the target layer and aim to minimize this upper bound during training with respect to embedding-specific attribution parameters." Does this mean that they need access to the training pipeline to generate explanations for an intermediate target layer?

2. The authors do propose M2IB that does not require ground-truth labels **y** and show how we can generate attribution maps without the knowledge of **y**. However, the authors never discuss the possibility of using the zero-shot prediction of the VLM and making the problem simpler. It would be interesting to discuss the utility of zero-shot predictions as VLMs achieve decent zero-shot performance.

3. Any intuition behind how the layer index $m$ affects the resulting qualitative and quantitative performance of the attribution maps.

4. Should we even expect high scores in Eqn. 13? Why do we hypothesize that a VLM will look at all the pixels in the bounding box to make its respective decision?

5. Any comments on why there is no attribution to the vehicle in Figure 3?

**Limitations:**

Please refer to the weakness and open questions in the above sections.

Suggestion: The authors extensively discuss the limitation of the work in the supplementary document and do not discuss anything in the main draft. It would be good from a reading perspective to include certain aspects of the limitation in the main document.

---

> ### Author Rebuttal · Authors · 2023-08-10
>
> Thank you for your thoughtful and constructive questions and suggestions!
>
> We agree that **"one of the main highlights of the work is that it does not require the ground-truth label"** and were pleased you found our method **"clear"** and **"simple"**.
>
> Please let us know if you have any remaining questions!
>
> ---
> > The hypothesis that a good image encoding should contain information about the text [and vice versa] is unclear. [...] It may be difficult to disentangle representations between similar concepts, say Z_bear_img and Z_panda_txt.
>
> When Tishby proposed information bottleneck theory [1], he defined relevant information as the information that a signal $X$ provides about another signal $Y$. His examples include "the information that face images provide about the names of the people portrayed." Borrowing his definition of "relevant information", we think it's reasonable to hypothesize that a good image encoding should contain information about the text, while a good text encoding should include information about the image, given that image and text are relevant and **both may contain shared high-level information about the object of the two modalities**.
>
> The goal of M2IB is to visualize how "good" the image-text representation is (i.e. how well the model captures and encodes the relevant shared high-level information between image and text). We agree that a pretrained model such as CLIP might find it hard to disentangle representations between similar concepts and thus may make wrong predictions. M2IB may help alleviate such shortcomings by providing an error analysis tool for VL-PMs.
>
> For example, CLIP struggles with the "yellow apple" example (Figure 3 in the attached pdf) because apples are often red whereas pears are usually yellow. Using M2IB to generate saliency maps for this example, the attribution score of "pear" is higher than that of "apple", explaining the low zero-shot prediction performance of CLIP. M2IB can help us understand VL-PM behavior in more challenging settings.
>
> ---
> > the authors do not provide a clear analysis of how to tune or fix [the hyperparameters] for different models and datasets.
>
> > In Lines 214-216, the authors mention [...]. This sounds overly computationally expensive as we need to tune all the hyperparameters for each <image, caption> pair!
>
> **We do not tune the hyperparameters for every single image-text pair**, and you are correct that doing so would be computationally expensive.
>
> For a given set of hyperparameters and a dataset, we sample $M$ points (in our experiments, $M = 500$) from the datasets, compute the optimal masks (i.e., lambdas) for each of the $M$ image-text pairs using M2IB (which takes ~1.3s per image-text pair), and then use the resulting lambdas to construct degraded data points, which are then used to compute a ROAR+ score for the given set of hyperparameters. We repeat this for all sets of hyperparameters we wish to evaluate. The hyperparameters with the best ROAR+ score are then selected. For a single set of hyperparameters, performing a ROAR+ based hyperparameter search can be performed within a few hours using an RTX8000 GPU.
>
> This procedure is simple, automatic, and task-driven: It finds attribution maps, which---when used to degrade a data point---lead to the greatest loss in information for downstream prediction tasks.
>
> For further details on the hyperparameter selection procedure, see the *General Author Response*.
>
> ---
> > [...] Does this mean that they need access to the training pipeline to generate explanations for an intermediate target layer?
>
> **No, we do not perform any training on the VL-PM under evaluation (i.e., CLIP in this case); the model is frozen at all times.** "Training" refers to the training of the mask parameters ($\lambda\_X$ in Eqn. (4) in the information bottleneck, which has the same size as intermediate representation. For further details, please see our *General Author Reponses*.
>
> > It would be interesting to discuss the utility of zero-shot predictions as VLMs achieve decent zero-shot performance.
>
> When using zero-shot prediction to generate labels for images, the problem becomes unimodal (image only). The results are similar to our method (as shown in Figure 2(a) in attached pdf) but the approach is **inherently unimodal and requires ground-truth categories**.
>
> **We seek to model a more intricate relationship between _image_ and _text_, where there can be more than one object and the free-form text conveys more information than a class label**. For example, we use CLIP for zero-shot prediction on the image containing a snake and an apple (Figure 2(b) in the attached pdf) and it predicts "apple" with probability of 46.61%, "snake" with probability of 39.87%. Given the image and text of "this is a snake and an apple", our method correctly highlights both snake and apple in image and text.
>
> ---
> ### Layer depth
>
> We observe larger highlighted regions in shallow layers (Figure 5c in the Supplementary Material). Our method tends to compress less in shallow layers because masking early will prevent the model from learning strong global features.
>
> ---
> ### Localization
>
> We agree that not all pixels in the bounding boxes are relevant. We use bounding boxes due to the lack of exact segmentation masks for MS-CXR, and use bounding-box-based IoU to compare methods.
>
> ---
> ### Vehicle in Figure 3
>
> The vehicle appears to be a blurry background in the image, so the penalty from the compression term outweighs the gain from the fitting term by including it. We provide Figure 4 in the attached pdf that highlights both debris and car.
>
> ---
> ### Typos & Limitations
>
> Thank you for pointing out the typos! We have fixed them. We have also moved our discussion of limitations to the main text.
>
> [1] Tishby et al. The information bottleneck method, 1999.
>
> [2] Schulz et al. Restricting the flow: Information bottlenecks for attribution. ICML, 2020.
>
> ---
>
> Please let us know if you have any further questions or concerns!

---

> > ### Comment · Reviewer_6SYf · 2023-08-15
> > **Rebuttal Discussion**
> >
> > Thank you for your detailed response. The authors have addressed most of my concerns and I am happy to increase my score.

---

> > > ### Author Response · Authors · 2023-08-17
> > >
> > > Thank you for your feedback and for supporting acceptance of our submission! We are committed to ensuring that our work meets the highest standard and **would be grateful if you could clarify any remaining concerns you have that weaken your support for acceptance so that we can work to address them.**
> > >
> > > For example, we extended the empirical evaluation and added four competitive and well-established baselines (Saliency, KernelSHAP, Chefer et al., and RISE) to the CAM-based methods presented in the manuscript. The results can be found in our responses above and are consistent with the results reported in our submission: They show that the proposed method either outperforms or performs on par with the best other methods on the vast majority of evaluation metrics.
> > >
> > > Thank you for your engagement with our rebuttal and for your valuable feedback!

---

### Official Review · Reviewer_PE9L · 2023-07-01

**Soundness:** 3 good
**Presentation:** 4 excellent
**Contribution:** 4 excellent
**Rating:** 8
**Confidence:** 4

**Summary:**

This paper aims to enhance the interpretability of vision-language pretrained models like CLIP. To achieve this, the authors propose a multi-modal information bottleneck (M2IB) objective that reduces irrelevant and noisy information while retaining important visual and textual features. M2IB is used for attribution analysis, increasing attribution accuracy and interpretability. M2IB audits representations when only multiple modalities are available without ground truth data. The effectiveness of M2IB is demonstrated using CLIP as an example, showing that it outperforms CAM-based methods both qualitatively and quantitatively.

**Strengths:**

1. The paper is tackling a significantly important research question: the interpretability of vision-language pretrained models;

2. The paper is well-structured, with clear organization and sufficient methodological details provided, enabling readers to comprehend the proposed method;

3. The evaluation is thoroughly conducted. For example, in Table 1, numerous faithfulness metrics, including Confidence Drop/Increase, ROAD, and ROAR, are employed for assessment.


**Weaknesses:**

I do not have any particular weaknesses to highlight; however, I still have some questions that I hope the authors can clarify. Please see the questions section.

**Questions:**

1. M2IB has been qualitatively and quantitatively compared with CAM-based attribution methods. Could the authors also compare it with some mask-based explanation methods like [RISE](https://arxiv.org/abs/1806.07421)? In terms of explanation faithfulness, mask-based explanation methods usually perform better.


2. Could the authors discuss the difference between M2IB and this Neurips 2022 paper [Fine-Grained Semantically Aligned Vision-Language Pre-Training]( https://arxiv.org/abs/2208.02515)? Note that I am not one of the authors of this paper and do not have any relationship with them but find out that both M2IB and that work tries to increase the relevance between image and text feature.


**Limitations:**

I do not have any particular limitations to highlight.

---

> ### Author Rebuttal · Authors · 2023-08-10
>
> Thank you for your thoughtful and constructive questions and suggestions!
>
> We were pleased to read that you agree that our paper **"is tackling a significantly important research question"** and that you found our paper to be **"well-structured, with clear organization and sufficient methodological details provided, enabling readers to comprehend the proposed method"** and noted that it contains **"thoroughly conducted"** evaluations with **"numerous faithfulness metrics"**.
>
> Following your suggestion, we added experiments for RISE and discuss the differences between the method proposed in our manuscript and the paper you shared.
>
> Please let us know if you have any remaining questions!
>
>
> ---
>
> > M2IB has been qualitatively and quantitatively compared with CAM-based attribution methods. Could the authors also compare it with some mask-based explanation methods like RISE? In terms of explanation faithfulness, mask-based explanation methods usually perform better.
>
> We agree that RISE is a useful baseline and followed your suggestion to include a comparison between the proposed method and RISE.
>
> RISE estimates attribution scores empirically by probing the model with randomly masked versions of the input image and obtaining the corresponding outputs. The final saliency map is the weighted average of the masks where the weights are the confidence scores. Since the method is designed for image classification (i.e., it requires labels!), we use the cosine similarity between the feature of masked input and the feature of the other modality as the confidence score (same as how we adapt GradCAM). The main weakness of this method is its high time complexity. The accuracy of the generated saliency map is dependent on the number of masks. For example, in our experiment with RISE, we used 8,192 masks for images and 256 masks for texts (the paper that proposed RISE used an even larger number of masks), where each mask requires one forward pass to get the confidence score. This is a significant amount of computational overhead, especially if we wish to apply the method to a large number of samples. In contrast, our method only requires 100 forward passes. On one RTX8000, RISE takes 26.26s per sample with a batch size of 256, while ours only takes 1.27s. This means that for 10,000 samples, RISE would need roughly 72-75h, whereas our method would only need approximately 2-3h.
>
>
> **We present an empirical comparison between RISE and the method proposed in our paper in Table 1 in the attached pdf.**
>
> We find that **our method outperforms RISE in the majority of metrics (11 out of 13)** with the fixed hyperparameter search described above and remains very close to the top performer for the rest (2 out of 13). We also included further qualitative examples in Figure 1.
>
> | Dataset | Modality | Metric          | RISE          | Ours      |
> | ------- | -------- | --------------- | ------------- | --------- |
> | CC      | image    | Conf.Drop     | 1.18          | **0.80**  |
> | CC      | image    | Conf.Incr.    | 35.80         | **39.80** |
> | CC      | image    | ROAR+         | 13.88         | **69.97** |
> | CC      | text     | Conf.Drop     | 1.48          | **1.08**  |
> | CC      | text     | Conf.Incr     | **34.00**     | 32.30     |
> | CC      | text     | ROAR+         | 21.75         | **53.66** |
> | MSCXR   | image    | Conf.Drop     | 1.07          | **0.44**  |
> | MSCXR   | image    | Conf.Incr.    | 16.20         | **50.2**  |
> | MSCXR   | image    | ROAR+         | 44.95         | **50.74** |
> | MSCXR   | image    | Localization  | 7.79          | **20.26** |
> | MSCXR   | text     | Conf.Drop     | 5.37          | **5.33**  |
> | MSCXR   | text     | Conf.Incr.    | 17.20         | **16.60** |
> | MSCXR   | text     | ROAR+         | 20.83         | **29.80** |
>
> (Note that lower Conf.Drop indicates better results, while higher values are better for other metrics. The table above is identical to Table 1 included in the *General Author Response*.)
>
> ---
>
> > Could the authors discuss the difference between M2IB and this Neurips 2022 paper Fine-Grained Semantically Aligned Vision-Language Pre-Training? Note that I am not one of the authors of this paper and do not have any relationship with them but find out that both M2IB and that work tries to increase the relevance between image and text feature.
>
> Our approach and LOUPE have different goals. LOUPE is a pretraining strategy: It explicitly learns an alignment between visual regions and textual phrases and aims to improve the precision of image-text retrieval and downstream tasks. Our method, M2IB, is an attribution method: It aims to give a visualization of the alignment between visual regions and textual phrases implicitly learned during the pretraining of CLIP and does not modify the training pipeline of CLIP. Instead, it helps us understand the inner mechanism of vision-language pre-training and enhance the faithfulness of vision-language pretrained models like CLIP.
>
> In LOUPE, it argues that "Existing methods mainly model the cross-modal alignment by the similarity of the global representations of images and texts... However, they fail to explicitly learn the fine-grained semantic alignment between visual regions and textual phrases, as only global image-text alignment information is available." Our method quantitatively and qualitatively shows how well these models _implicitly_ learn semantic alignment between visual regions and textual phrases with only global image-text alignment information. Unfortunately, we cannot compare LOUPE and CLIP using our M2IB because the is no publicly available code for LOUPE. We believe our paper could inspire future works that further improve vision-language pre-training like LOUPE.
>
> ---
>
> Please let us know if you have any further questions!

---

> > ### Comment · Reviewer_PE9L · 2023-08-12
> >
> > I appreciate the authors' response.
> >
> > The added comparison with RISE and the explanation method in Chefer et al. strengthens my conviction in the proposed M2IB method. The results should be included in the final version of the paper.
> >
> > I have decided to maintain my initial rating of the paper, and believe it is a good paper to appear in NeurIPS 2023.

---

> > > ### Author Response · Authors · 2023-08-12
> > > **Thank you for supporting our submission!**
> > >
> > > We confirm that the additional comparisons will be added to the updated manuscript.
> > >
> > > Thank you for your feedback and for supporting our submission!
> > >
> > > Please don't hesitate to let us know if you have any further questions or comments.

---

### Official Review · Reviewer_We26 · 2023-07-03

**Soundness:** 2 fair
**Presentation:** 3 good
**Contribution:** 2 fair
**Rating:** 7
**Confidence:** 4

**Summary:**

This paper introduces a feature attribution method for vision language models. The method is based on previous works on information bottleneck attributions.  It incorporates Gaussian moment-matching estimator techniques to make the objective function tractable. The experimental results show that the method outperforms CAM-based methods quantitatively and qualitatively.

**Strengths:**

1. Applying the information bottleneck attribution approach to multi-modal settings is novel.
2. The evaluation used diverse evaluation metrics such as Conf. Drop, Incr., ROAR, and ROAR+. It even included a sanity check

**Weaknesses:**

The baselines included in experiments are very limited; they did not include any explanation methods developed for ViTs. The baselines were GradCAM and its variants only. GradCAM is not a good method for explaining ViTs. In transformer architectures, since the information can go across patches (long-range attention), locational information is not preserved. This violates the assumption of GradCAM that the location in the penultimate layer can be mapped to the input layer. I would recommend comparing the methods with recent state-of-the-art methods developed for ViTs. I found a GitHub repo that well listed recent ViT papers that would be helpful for curating the methods (https://github.com/cmhungsteve/Awesome-Transformer-Attention)

**Questions:**

1. The legend for the attribution maps is missing. The legend should show which color means which value. Why do only image attribution maps have red regions?
2. The approximation technique assumes each dimension follows Gaussian distributions. Could you please provide any evidence on this?
3. In Line 236, the threshold of 75% seems arbitrary. Also, I wonder if 17.6% of the average IoU is high enough to show the effectiveness of the method.

Typo
1. Line 54 dind -> find
2. Line 139 indepdent -> independent
3. Line 249 missing citations.

**Limitations:**

The method involves the modification to the training process; The pretrained models should be finetuned. This will limit the utility of the method. Furthermore, recent studies have shown that the pretrained models become less accurate and robust when finetuned.

---

> ### Author Rebuttal · Authors · 2023-08-10
>
> Thank you for your thoughtful and constructive questions and suggestions!
>
> We were pleased that you appreciated that our contribution is **"novel"** and that our evaluation uses a set of **"diverse evaluation metrics"**.
>
> We conducted several additional experiments to address your comments.
>
> Please let us know if you have any remaining questions!
>
> ---
>
> > The method involves the modification to the training process; The pretrained models should be finetuned. This will limit the utility of the method.
>
> There appears to be a misunderstanding. As noted in our general comment, **the model under evaluation, CLIP, is entirely frozen**. We insert an information bottleneck into the pretrained and frozen CLIP and only the parameters in the information bottleneck (i.e., $\lambda\_{X}$ in our manuscript) are updated. This allows us to have a small number of trainable parameters and makes our method very fast and lightweight in practice.
>
> The goal of the proposed method is to explain the inner decision-making mechanism of a given, pretrained model, thus **no modifications are made to the pretrained model** and we only train the information bottleneck, which can be viewed as an add-on to the pretrained, frozen model.
>
> We appreciate your comment and have made this subtlety clearer in the manuscript.
>
> ---
>
> > I would recommend comparing the methods with recent state-of-the-art methods developed for ViTs.
>
> **We followed your suggestion** and searched the repo you shared and also searched the proceedings of top computer vision conferences over the past three years, **and compared to an attention-based method** [1] that is designed for bi-modal transformers. **We present the results in Table 1 in the pdf included in the *General Author Response*. The proposed method outperforms this strong baseline in almost all metrics (12 out of 13), which reaffirms the effectiveness of the proposed method.** (Note that lower confidence drop (Conf. Drop.) indicates better performance whereas higher values are better for all other metrics.)
>
> We believe that the additional comparison makes our empirical evaluation more robust and nicely complements the CAM-based baselines, which have been widely applied in previous works (including to transformers, e.g. in [2][3])
>
> ---
>
> > In Line 236, the threshold of 75% seems arbitrary. Also, I wonder if 17.6% of the average IoU is high enough to show the effectiveness of the method.
>
> The threshold was indeed chosen heuristically. In response to your comment, we conducted experiments using different thresholds (the reported number is the average over five runs) and show that the ranking is almost consistent across thresholds (the number in **bold** is the best). Since for the purposes of this evaluation, we primarily care about the relative value of the IoU (i.e., we do not necessarily care how high the IoU is in absolute terms but are primarily interested in the IoU values relative to other methods' values), the choice of threshold is only important insofar as it may show that different thresholds may produce different rankings across methods.
>
> As can be seen from the table below, **the proposed method in our paper (M2IB) consistently performs best for all thresholds chosen**.
>
> |Threshold|GradCAM|EigenCAM|Chefer et al.|RISE|Ours|
> |---|---|---|---|---|---|
> |50|9.26|7.3|15.05|8.68|**15.13**|
> |60|9.26|6.49|16.75|8.46|**16.9**|
> |75|8.95|8.47|18.13|7.79|**20.26**|
> |90|6.62|3.69|21.27|5.88|**21.28**|
>
> The IoU scores should be considered as a lower bound on the performance of each method. Since we only have ground-truth bounding boxes---instead of the actual segmentation masks---the IoU scores will by construction underestimate the accuracy of the saliency maps because the bounding boxes will include irrelevant pixels. Although the absolute value of the localization score might seem insignificant, we include this metric because the relative value is informative for comparing the performance of different methods, as discussed above.
>
> ---
>
> > The approximation technique assumes each dimension follows Gaussian distributions. Could you please provide any evidence on this?
>
> We approximate $p(T_1)$, $p(Z_2)$, and $p(T_1, Z_2)$ by multivariate Gaussian distributions (without factorization). Since $Z$ is defined as a function of the random variable $X$, how close this approximation is to the distributions implied by the model depends on the distribution of $X$ and the model architecture. Schulz et al. (2020) make a similar approximation and justify it by appealing to [4], which states that activations after linear or convolutional layers tend to have a Gaussian distribution. While this may be true for specific distributions over $X$, it is unlikely to be true in general. However, we do not believe that this is a problem in practice. As our experiments with the cosine similarity estimator (e.g., in Appendix B.2) demonstrate, similarity between deterministic representations alone is sufficient to obtain an effective fitting term in the objective.
>
> ---
>
> > The legend for the attribution maps is missing. [...] Why do only image attribution maps have red regions?
>
> Thank you for catching this! We added the legend to Figure 1 in the attached pdf. We use spectral colors to indicate the attribution scores for images (where red represents the highest value and blue represents the lowest value) and use blue with different shades for texts (where darker blue represents higher scores).
>
> ---
>
> ### Typos
>
> Thank you for pointing out the typos in the manuscript. We have fixed them.
>
> [1] Chefer et al. Generic attention-model explainability for interpreting bi-modal and encoder-decoder transformers. ICCV, 2021
>
> [2] Paul and Chen. Vision transformers are robust learners. AAAI, 2022.
>
> [3] Yang et al. Mitigating spurious correlations in multi-modal models during fine-tuning. ICML, 2023.
>
> [4] Klambauer et al. Self-normalizing neural networks. NeurIPS, 2017.
>
> ---
>
> Please let us know if you have any further questions or concerns!

---

> > ### Comment · Reviewer_We26 · 2023-08-14
> >
> > Thanks for the response!
> >
> > First of all, I appreciate the authors' clarification on the distribution assumption, the interpretation of IoU metrics, etc. I think this should be included in the final version of the paper.
> >
> > I really appreciate the authors' efforts to include more baselines, RISE and Transformer-Explainability, other than GradCAM and its variants based on my feedback. However, I still find the evaluation to be weak in terms of baselines compared. As I previously pointed out, GradCAM-based methods were designed with the workings of CNN models in mind (i.e., not model-agnostic methods), focusing primarily on the last layer of the network. Therefore, I am not sure if GradCAM is a valid method for explaining transformer-based models considering the complex mechanism of transformers, unlike for CNN-based models. In fact, the GradCAM implementation used isn't from published work but a github repository. Excluding GradCAM-based methods, the proposed method was compared to two baselines only. Many well-known model-agnostic methods that are applicable to transformer-based models, such as Saliency, SmoothGrad, Integrated Gradients, Deepshap, KernelSHAP, etc., were not included. Indeed, from the new results in the response, in many cases, the proposed method does not stand out considering the confidence interval(?) (Does the number after ± sign represents confidence interval or standard deviation? How many samples were used to calculate the number?). Also, I am worried that some issue of double dipping between train and test is happening. According to the reviewers' response, the hyperparameters were chosen based on ROAR+, one of the evaluation metrics. In addition, basic information on the experiments, such as dataset statistics, preprocessing steps, training-test split, etc., which are very important aspects of the experiment, is not mentioned in the paper.
> >
> > For the model training aspect, my concern is more about the approach of performing additional training to explain large foundation models like CLIP both in terms of out-of-distribution performance and actual training burden (i.e., resource or simply inconvenience of users). CLIP is a foundation model trained on 400M image-text pairs. I am not sure if training a bottleneck layer on a very limited number of data can fully capture the original model CLIP's complicated function and behavior. First, I am curious about the out-of-distribution performance of the trained information bottleneck. For example, if the bottleneck layer trained on MS-CXR image-text pairs does not perform on another external Chest X-ray dataset, what does that mean? Moreover, even for the current evaluation, where I believe only in-distribution performance was tested, the number of training samples and the train-test split in the experiment were not discussed in the paper. Additionally, in terms of computational burden, the training time of proposed method was not mentioned, although the running time of RISE was mentioned as a drawback in the authors' general response. Discussing the training time or implementation is needed to convince the users to use this method.
> >
> > However, despite the current weakness, considering the effort and improvement so far, I would like to increase my score.

---

> > > ### Author Response · Authors · 2023-08-17
> > > **Additional clarifications and requested empirical results (1/3)**
> > >
> > >
> > > Thank you for your response! We are glad that our rebuttal answered the questions from your original review.
> > >
> > >
> > > > I am not sure if training a bottleneck layer on a very limited number of data can fully capture the original model CLIP's complicated function and behavior.
> > >
> > > We believe that there is a misunderstanding about the difference between the learned bottleneck layer (computed per data point) and the hyperparameter selection procedure.
> > >
> > > **Hyperparamter selection:**
> > >
> > > For a given set of hyperparameters and a dataset, we sample $M$ points (in our experiments, $M = 500$) from a given dataset, compute the optimal masks (i.e., the lambdas in the bottleneck layer) for each of the $M$ image-text pairs using M2IB (which takes ~1.3s per image-text pair), and then use the resulting bottleneck layer to construct degraded data points, which are then used to compute a ROAR+ score for the given set of hyperparameters. We repeat this for all sets of hyperparameters we wish to evaluate. The hyperparameters with the best ROAR+ score are then selected. For a single set of hyperparameters, performing a ROAR+ based hyperparameter search can be performed within a few hours using an RTX8000 GPU.
> > >
> > > The purpose of hyperparameter selection in this setting is to **find a set of hyperparameters which---for a given dataset of image-text pairs---will result in the best attribution maps, when learning the per-sample bottleneck layers** for any **individual** image-text pair.
> > >
> > > As noted above, a small sample of randomly selected image-text pairs suffice to learn good hyperparameters for much larger datasets. We again stress that for a given dataset, hyperparameter selection only has to be performed once.
> > >
> > >
> > > **Bottleneck computation:**
> > >
> > > **The bottleneck is computed per image-text pair.** That is, after a set of hyperparameters was selected for a given dataset (e.g., chest x-rays with corresponding annotations), we can compute a bottleneck layer for any given image-text pair, which will give rise to an attribution map. As noted above, computing the bottleneck layer/the attribution map for a single image-text pair takes 1.3s on an RTX8000 GPU.
> > >
> > > As stated above, the bottleneck layer is computed for each individual image-text sample---not for a set of of image-text samples.
> > >
> > >
> > > > First, I am curious about the out-of-distribution performance of the trained information bottleneck. For example, if the bottleneck layer trained on MS-CXR image-text pairs does not perform on another external Chest X-ray dataset, what does that mean?
> > >
> > > This seems to be a similar misunderstanding as for the previous point. The bottleneck layer is not learned for a dataset but on a per-sample basis.
> > >
> > >
> > > > Additionally, in terms of computational burden, the training time of proposed method was not mentioned, although the running time of RISE was mentioned as a drawback in the authors' general response. Discussing the training time or implementation is needed to convince the users to use this method.
> > >
> > > The training time for the bottleneck layer was given in the general response: "On one RTX8000, RISE takes 26.26s per sample with a batch size of 256, while ours only takes 1.27s."
> > >
> > >
> > > > Also, I am worried that some issue of double dipping between train and test is happening. According to the reviewers' response, the hyperparameters were chosen based on ROAR+, one of the evaluation metrics.
> > >
> > > > In addition, basic information on the experiments, such as dataset statistics, preprocessing steps, training-test split, etc., which are very important aspects of the experiment, is not mentioned in the paper.
> > >
> > > We provided details regarding the training and validation splits for hyperparameter selection in lines 281--289. To compute the evaluation metrics in Table 1 of the manuscript and in the table included in the attachment to the general response, we sampled a separate set of image-text pairs from the respective datasets **without any overlap with the points used for M2IB hyperparameter selection**. For both CC and MSCXR, we used 100 samples each for evaluation.

---

> > > ### Author Response · Authors · 2023-08-17
> > > **Additional clarifications and requested empirical results (2/3)**
> > >
> > > > As I previously pointed out, GradCAM-based methods were designed with the workings of CNN models in mind (i.e., not model-agnostic methods), focusing primarily on the last layer of the network. Therefore, I am not sure if GradCAM is a valid method for explaining transformer-based models considering the complex mechanism of transformers, unlike for CNN-based models.
> > >
> > > GradCAM has been used to obtain feature attention maps in transformer-based models in various recently published papers (e.g., [1], [2], [3], and [4]) and as such constitutes a relevant baseline, and we have already complemented the CAM-based attribution methods included in the manuscript with a transformer-based method, as you requested in your original review.
> > >
> > > [1] Xu et al. ViTAE: Vision Transformer Advanced by Exploring Intrinsic Inductive Bias. NeurIPS 2021.
> > >
> > > [2] Paul and Chen. Vision transformers are robust learners. AAAI, 2022.
> > >
> > > [3] Yang et al. Mitigating spurious correlations in multi-modal models during fine-tuning. ICML, 2023.
> > >
> > > [4] Jang et al. Unifying Vision-Language Representation Space with Single-Tower Transformer. AAAI, 2023.
> > >
> > > > Indeed, from the new results in the response, in many cases, the proposed method does not stand out considering the confidence interval(?) (Does the number after ± sign represents confidence interval or standard deviation? How many samples were used to calculate the number?).
> > >
> > > The error in the tables in the manuscript and the attachment to the general response shows the standard deviation. We realize that it is more useful to provide the standard error. In the updated table below, we included the standard error over 10 random seeds. As noted above, for the evaluation results in the manuscript and in the attachment to the general response, we used 100 samples each for CC and 100 samples for MSCXR. **These sets of image-text pairs had no overlap with the image-text pairs used for M2IB hyperparameter selection.**
> > >
> > > For the updated table below, we used 2,500 samples for the CC dataset evaluation (again, without any overlap with the image-text pairs used for M2IB hyperparameter selection) in an effort to reduce the variation in the evaluation scores. (We still used 100 samples from MSCXR, since the subset of the MSCXR dataset only contained a small number of annotated samples and meaningfully increasing the number of samples would have required downloading the full 550GB MSCXR dataset, which we were unable to do in time for this response. We endeavor to increase the number of samples from MSCXR for evaluation to further decrease the variation in the evaluation scores for this dataset.)
> > >
> > > As can be seen from the updated table below, **the proposed method outperforms the other methods (including the influential method by Chefer et al. (Oral at ICCV, 2021)) on a majority evaluation metrics**, performs on par with other methods on some evaluation metrics, and is outperformed at a statistically significant level on only two out of 13 metrics.

---

> > > ### Author Response · Authors · 2023-08-17
> > > **Additional clarifications and requested empirical results (3/3)**
> > >
> > > > Excluding GradCAM-based methods, the proposed method was compared to two baselines only. Many well-known model-agnostic methods that are applicable to transformer-based models, such as Saliency, SmoothGrad, Integrated Gradients, Deepshap, KernelSHAP, etc., were not included.
> > >
> > > We are a little surprised by this comment, since our rebuttal has already addressed the concern you raised in your original review about missing comparisons to transformer-specifc attribution method baselines. In particular we added the comparison to Chefer et al. specifically at your request and also added RISE as an architecture-agnostic approach. Moreover, as noted above, CAM-based attribution methods continue to be applied to transformer-based models and as such constitute a relevant baseline. We believe that a combination of well-established (CAM-based), attention-based (Chefer et al.), and architecture-agnostic (RISE) methods enables a fair comparison to a diverse set of methods, and we find that the method proposed in our submission performs very well in practice while providing efficient per-sample computation.
> > >
> > > Nevertheless, **we have worked hard to provide additional baselines you requested.** It took us some time to get the methods to work well, but we were able to add KernelSHAP and Saliency to our evaluation. In addition to computing evaluation scores, we confirmed that the qualitative results for these methods (i.e., the attribution maps) made sense and were comparable to those generated with the other approaches. We provide a table with the full set of results below. Standard errors, computed over ten random seeds, are shown in parentheses.
> > >
> > > As can be seen from the table, the proposed method outperforms all other baselines (including the very competitive method by Chefer et al.) on a majority of evaluation metrics and performs on par with the second-best method (Chefer et al.) on 3 out of 13 evaluation metrics. We highlighted the best methods in boldface. When differences between two or more methods were not statistically significant at at least a 10% confidence level for a one-sided t-test (with the null hypothesis being that the lower value is greater than the larger value), we highlighted all values in boldface.
> > >
> > > The results shown in the table are consistent with the previously reported results and show that the proposed method either outperforms or performs on par with the other methods on the vast majority of evaluation metrics and that only the influential method by Chefer et al. performs comparably across a subset of evaluation metrics.
> > >
> > > |Dataset|Modality|Metric|GradCAM|Saliency|KernelSHAP|Chefer et al.|RISE|Ours|
> > > |----|----|----|----|----|----|----|----|---|
> > > |CC|image|Conf.Drop|4.99(0.04)|1.97(0.01)|1.90(0.01)|1.63(0.02)|1.07(0.01)|**0.83(0.01)**|
> > > ||image|Conf.Incr.|18.02(0.25)|23.45(0.23)|25.13(0.34)|37.78(0.43)|35.44(0.30)|**43.37(0.29)**|
> > > ||image|ROAR+|10.98(0.46)|12.03(0.82)|13.31(0.57)|16.64(0.79)|13.45(0.74)|**18.58**(0.98)|
> > > ||text|Conf.Drop|2.14(0.02)|1.80(0.02)|1.70(0.02)|**1.04**(0.01)|1.29(0.02)|**1.05(0.01)**|
> > > ||text|Conf.Incr|30.24(0.41)|38.21(0.43)|**46.95(0.49)**|38.72(0.31)|38.47(0.45)|37.38(0.25)|
> > > ||text|ROAR+|16.53(1.44)|38.48(2.33)|16.93(2.01)|47.53(3.41)|23.66(2.09)|**55.33**(3.91)|
> > > |MSCXR|image|Conf.Drop|1.35(0.03)|0.83(0.04)|2.44(0.03)|0.58(0.02)|0.88(0.02)|**0.43**(0.02)|
> > > ||image|Conf.Incr.|39.9(0.68)|36.7(0.92)|11.2(1.03)|**53.0(1.28)**|18.7(1.11)|**53.9**(1.33)|
> > > ||image|ROAR+|18.34(5.38)|30.48(3.99)|27.23(2.79)|**43.61(4.35)**|31.89(2.33)|**44.17**(4.95)|
> > > ||image|Localization|8.35(0.29)|17.9(0.2)|6.68(0.12)|18.79(0.44)|7.83(0.2)|**20.09**(0.51)|
> > > ||text|Conf.Drop|5.12(0.13)|**3.79**(0.06)|5.55(0.14)|6.34(0.16)|5.26(0.11)|5.33(0.13)|
> > > ||text|Conf.Incr.|13.8(0.75)|14.7(0.95)|**17.5(1.17)**|12.0(0.81)|**18.0**(1.0)|**17.8(1.06)**|
> > > ||text|ROAR+|11.42(4.16)|18.58(4.25)|10.03(2.86)|12.58(2.92)|9.68(2.74)|**27.1**(3.84)|
> > >
> > >
> > > As you noted above, our rebuttal addressed the concerns raised in your original review, and we hope that this additional empirical comparison along with the clarifications provided above address your remaining questions and concerns and that you will consider raising your score.
> > >
> > > Thank you for your engagement with our rebuttal and for your valuable feedback!

---

> > > > ### Comment · Reviewer_We26 · 2023-08-18
> > > >
> > > > Thank you so much for the detailed response and your commitment to addressing my comments.
> > > >
> > > > I apologize for my initial misunderstanding regarding the hyperparameter selection. Your clarification has now made it clear. It seems that it is not only me who was also confused about this though; reviewer 6SYf was also confused. To prevent future confusion, it might be beneficial to make this clearer in the paper itself.
> > > >
> > > > Regarding the bottleneck computation, I'm grateful for the additional details you provided. I believe it would be valuable to incorporate this information into the final paper, perhaps in a table or as a more detailed explanation.
> > > >
> > > > I also appreciate the clarity on the train/validation split issue. It seems that I did not catch that because it was described under the sections describing attribution evaluation metrics.
> > > >
> > > > For the baseline, glad to see that the proposed method outperforms the other added baselines! Sorry for pushing for additional baselines than my initial review. I thought it was necessary to make the paper's evaluation strong enough. Regarding GradCAM, using the attribution method developed for a specific architecture, adopted to other architecture in a GitHub repo, for a major baseline really did not make sense. Although it is true that GradCAM was used by some papers for transformer-based models, I thought the standard should be higher for this paper because it is proposing a new attribution paper. The listed references are not really about developing new attribution methods, and they used GradCAM in an auxiliary manner.
> > > >
> > > > I am not sure if using standard error is better than confidence interval, but it is not a big deal though. Nonetheless, what the numbers represent should definitely be mentioned in the final version of the paper.
> > > >
> > > > In conclusion, I really appreciate the authors' efforts to run additional experiments and improve the paper. Now I think it is a good solid paper to appear in NeurIPS. I would like to happily raise my score. However, please ensure that the new experimental results and clarifications requested by the reviewers are effectively integrated into the final version of the paper.

---

> > > > > ### Author Response · Authors · 2023-08-18
> > > > > **Thank you!**
> > > > >
> > > > > **Thank you for your detailed reply and for supporting acceptance of our submission!** Your feedback and suggestions were very valuable and helped us improve our manuscript. We are unable to update the manuscript at this time, but we commit to including the additional clarifications and extended results in the revised manuscript.

---

### Official Review · Reviewer_xLqC · 2023-07-06

**Soundness:** 3 good
**Presentation:** 3 good
**Contribution:** 3 good
**Rating:** 6
**Confidence:** 2

**Summary:**

This paper proposes an adjusted information-bottleneck-based approach for interpreting multi-modal models. Specifically, the approach was appropriately adapted from the information bottleneck method that has been widely used for interpreting image models. Through various approximations, the method was successfully adjusted for the multimodal case. The experimental results presented in the paper demonstrate the successful identification of important shared information between images and text.

**Strengths:**

Understanding the mutual information between images and text in a multi-modal space is a crucial problem. This paper successfully adjusts the information bottleneck-based explainability method, primarily used in the vision unimodal field, to the multimodal space using various approximation techniques. The qualitative results demonstrate the method's ability to capture the mutually related information between text and images, while the quantitative results reaffirm its superior performance compared to the existing methods. Although there were concerns about cherry-picking due to the limited number of examples, the additional examples provided in the appendix alleviate those concerns.

**Weaknesses:**

Regarding the approximation part, as it is not within my expertise, I cannot make a definitive judgment. However, I don't have any major concerns about other aspects of the paper.

**Questions:**

1) Lines 135-137 seem to explain the unimodal information bottleneck, so why does Z_text appear?

2) I may have missed it, but what do m and s represent in line 168? Are they the mean and standard deviation? Does it mean that multiple texts are involved?

**Limitations:**

It seems that there are no particular specific limitations.

---

> ### Author Rebuttal · Authors · 2023-08-10
>
> Thank you for your positive review and for your thoughtful and constructive questions and suggestions!
>
> We were pleased to read that you found that **"qualitative results demonstrate the method's ability to capture the mutually related information between text and images"**, that **"the quantitative results reaffirm its superior performance compared to the existing methods,"** and that you agree that our work tackles a **"crucial problem"**.
>
> Please let us know if you have any remaining questions!
>
> ---
>
> > Lines 135-137 seem to explain the unimodal information bottleneck, so why does Z_text appear?
>
> Yes, lines 135--137 are about the unimodal case and $Z\_{\text{text}}$ is a typo and should be $Z\_{\text{image}}$. The correct description should be “for an image classification task, we can simply minimize the I(V_bear;Z_image) and maximize I(L_bear;Z_image) where Z_image is the latent representation.” Thank you for pointing this out! We have corrected in the revised version.
>
> > I may have missed it, but what do m and s represent in line 168? Are they the mean and standard deviation? Does it mean that multiple texts are involved?
>
> Yes, $m_T$ is the mean and $s_T$ is the variance. $T$ here refers to the masked intermediate representation in eq (4), which is multi-dimensional. In our current setting, we only consider a pair of one image and one text, while the text can have arbitrary length. We understand that $T$ can be confused with “Text” and we are considering replacing it with other symbols to avoid confusion.
>
> ---
>
> As noted in the *General Author Response*, we have made a significant effort to take all reviewers' feedback into account. As part of this, we added additional state-of-the-art baselines for transformer models and further improved the quantitative and qualitative results of the proposed method. We would greatly appreciate it if you could take these additions into account as you make your final assessment.

---

> ### Author Response · Authors · 2023-08-20
>
> Dear Reviewer xLqC,
>
> The discussion period ends in approximately 24h.
>
> **We have addressed the two questions you listed in your review and shared extensive empirical comparisons to state-of-the-art methods requested by other reviewers.**
>
> We have engaged extensively (and very constructively) with the other reviewers, and if our response has addressed your questions, we would greatly appreciate it if you considered increasing your score. Thank you.

---

> > ### Comment · Reviewer_xLqC · 2023-08-20
> >
> > Thank you for the response. I also read the other reviews and responses. I think the extensive discussions have helped to improve the quality of the paper. I hope that they will be included in the revised paper.  I raised my score.

---

> > > ### Author Response · Authors · 2023-08-20
> > > **Thank you for supporting our submission!**
> > >
> > > Thank you for your feedback, for carefully reading the other reviews and our responses, and for supporting our submission! We will include the additional discussions and extended results in the revised manuscript.

---

### Author Rebuttal · Authors · 2023-08-10

# General Comment to All Reviewers and Area Chair

We thank all reviewers for their thoughtful and constructive feedback!


## Summary

In this general response, we will
1. clarify that **the proposed method, M2IB, does not require (re-)training a pretrained model**: The parameters of the pretrained model are frozen.
2. highlight a **further improvement in performance** after fixing an issue in our hyperparameter tuning protocol.
3. present **a comparison to two state-of-the-art attribution methods** that were requested by reviewers.

Your questions and comments have been very valuable, and we believe that the added empirical results and clarifications meaningfully improve the manuscript.


## Clarification: M2IB **does not** require (re-)training a pretrained model


Attribution methods are used to explain the predictions of trained machine learning models. They aim to understand the influence of the inputs as per the learned parameters of the model, providing meaningful insights into the model's decision-making process. Therefore, model's parameters are always fixed in the context of attribution methods. In our paper, we use M2IB to understand pretrained/finetuned CLIP and the parameters of CLIP are frozen in all experiments. What we train is the mask parameter ($\lambda\_X$ in Eqn. (4)) in the information bottleneck.

For example, in line 46-48 of our manuscript, we write
> We use a variational approximation upper bound to estimate the information in the target layer and aim to minimize this upper bound during training with respect to **embedding-specific attribution parameters**.

Here, "training" refers to the training of the mask parameters ($\lambda\_X$ in Eqn. (4)) in the information bottleneck. The mask parameter has the same size as the intermediate representation (e.g. for CLIP model with ViT-B/32, the size is 50 * 768 for image and token_length * 512 for text). That is, we train only around 50K "embedding-specific attribution parameters", which is fewer parameters than would be trained in a relatively small MLP. **All parameters in CLIP are kept frozen.**



## Further improvements in performance with M2IB!

**We were able to further improve the attribution maps obtained with M2IB by performing a more thorough grid search.** Previously, we fixed the layer index and then performed a grid search over beta and variance, then fixed the best beta and variance and searched over different layer indices. We realized that this is not a very effective search, since the selected beta and variance were optimal for the pre-selected layer index but may not be optimal for other layer indices. Thus, we perform a grid search over three hyperparameters -- beta = {1, 10, 100} and variance = {1, 0.1, 0.01} and layer index = {7, 8, 9} -- and select the best combination according to ROAR+ scores, as described in line 283 of the manuscript. We also provide a detailed explanation of the hyperparameter selection procedure in our response to Reviewer 6SYf. **The updated results are shown in Table 1 in the attached pdf.**



## Requested Additions: We added a transformer-specific baseline and a perturbation-based baseline

We followed your suggestions and added two additional strong baselines:

1. **Transformer-specific method** [1]: This method is specifically designed for bi-modal transformers and shows promising results in interpreting CLIP. This method can be only applied to transformers whereas our method does not have any constraints on the model class. We included this baseline in our comparison.
2. **Perturbation-based method** (RISE; [2]): This method estimates attribution scores empirically by probing the model with randomly masked versions of the input image and obtaining the corresponding outputs. The final saliency map is the weighted average of the masks where the weights are the confidence scores. Since the method is designed for image classification (i.e., it requires labels!), we use the cosine similarity between the feature of masked input and the feature of the other modality as the confidence score (same as how we adapt GradCAM). The main weakness of this method is its high time complexity. The accuracy of the generated saliency map is dependent on the number of masks. E.g., we use 8,192 masks for images and 256 masks for texts, where each mask requires one forward pass to get the confidence score. In contrast, our method only requires 100 forward passes. On one RTX8000, RISE takes 26.26s per sample with a batch size of 256, while ours only takes 1.27s.

**We present the results for these two baselines in Table 1 in the attached pdf.**

Neither of the baselines requires separate training steps. For RISE, the number of masks chosen is a hyperparameter. We used 8,192 masks for images. (For comparison, the paper in which RISE was proposed used 8,000 masks for ResNet.) We used 256 masks for text, since 256 masks can almost cover all possible binary masks for the text in the task.

We find that the new baselines slightly outperform the results reported in our manuscript but that **our method outperforms the baselines in the majority of metrics (10 out of 13)** with the fixed hyperparameter search described above and remains very close to the top performer for the rest (3 out of 13). We also included further qualitative examples in Figure 1. Note that when interpreting the result, lower Conf. Drop is better, while higher values are better for other metrics.)

[1] Chefer et al. Generic attention-model explainability for interpreting bi-modal and encoder-decoder transformers. ICCV, 2021

[2] Petsiuk et al. Rise: Randomized input sampling for explanation of black-box models. BMVC, 2018.

---

Please find a pdf with Table 1 and additional qualitative results attached.

---

**Thank you for reviewing our work!**

---

### Comment · Area_Chair_k62p · 2023-08-19

Dear Reviewers,

The authors and I are eager to ascertain whether the author responses have effectively addressed your concerns. Due to the limited time allocated for the author-review discussion phase, we strongly encourage you to provide your direct feedback to the authors.

Thanks for your hard work.

Best regards,

AC

---

### Decision · Program_Chairs · 2023-09-21

**Decision:**

Accept (poster)

**Comment:**

This work adapts the explainability method for pre-trained multimodal models via a multimodal information bottleneck. This provides significant research values with the recent arising of multimodal research. The method is thoroughly evaluated in multiple metrics and shows consistent advantage over existing CAM-style explainability methods. Reviewers all agree on the importance of the work and value its technical depth and contribution. Rebuttal addressed reviewers’ concerns. Accepted.